

# DRYP 1.0: A parsimonious hydrological model of DRYland Partitioning of the water balance

Edisson A. Quichimbo [1], Michael Bliss Singer [1,3,4], Katerina Michaelides [2,4,5], Daniel E.J. Hobley [1], Rafael Rosolem [5,6], Mark O. Cuthbert [1,3,7]

[1]School of Earth and Environmental Sciences, Cardiff University, Cardiff, CF10 3AT, UK
[2]School of Geographical Sciences, University of Bristol, Bristol, BS8 1SS, UK
[3]Water Research Institute, Cardiff University, Cardiff, CF10 3AT, UK
[4]Earth Research Institute, University of California Santa Barbara, Santa Barbara, California, USA
[5]Cabot Institute for the Environment, University of Bristol, Bristol, UK
[6]University of Bristol, Faculty of Engineering, University Walk, Clifton BS8 1TR, UK
[7] School of Civil and Environmental Engineering, The University of New South Wales, Sydney, New South Wales, Australia

*Correspondence to*: Edisson A. Quichimbo (quichimbomiguitamaea@cardiff.ac.uk)

**Abstract.** Dryland regions are characterized by water scarcity and are facing major challenges under climate change. One difficulty is anticipating how rainfall will be partitioned into evaporative losses, groundwater, soil moisture and runoff (the water balance) in the future, which has important implications for water resources and dryland ecosystems. However, in order to effectively estimate the water balance, hydrological models in drylands need to capture the key processes at the appropriate spatiotemporal scales including spatially restricted and temporally brief rainfall, high evaporation rates, transmission losses and focused groundwater recharge. Lack of available data and the high computational costs of explicit representation of ephemeral surface-groundwater interactions restrict the usefulness of most hydrological models in these environments. Therefore, here we have developed a parsimonious hydrological model (DRYP) that incorporates the key processes of water partitioning in dryland regions, and we tested it in the data-rich Walnut Gulch Experimental Watershed against measurements of streamflow, soil moisture and evapotranspiration. Overall, DRYP showed skill in quantifying the main components of the dryland water balance including monthly observations of streamflow (Nash efficiency (NSE) ~0.7), evapotranspiration (NSE >0.6) and soil moisture (NSE ~0.7). The model showed that evapotranspiration consumes > 90 % of the total precipitation input to the catchment, and that <1 % leaves the catchment as streamflow. Greater than 90 % of the overland flow generated in the catchment is lost through ephemeral channels as transmission losses. However, only ~35 % of the total transmission losses percolate to the groundwater aquifer as focused groundwater recharge, whereas the rest is lost to the atmosphere as riparian evapotranspiration. Overall, DRYP is a modular, versatile and parsimonious Python-based model which can be used to anticipate and plan for climatic and anthropogenic changes to water fluxes and storage in dryland regions.

## 1. Introduction

Drylands are regions where potential evapotranspiration far exceeds precipitation and where water is scarce. Consequently, the water balance in such area is highly sensitive to climatic forcing in terms of the delivery of precipitation and the evaporative demand from the atmosphere (Goodrich et al., 1997; Kipkemoi et al., 2021; Pilgrim et al., 1988; Zoccatelli et al., 2019). A key challenge is anticipating how rainfall partitioning into evaporative losses, groundwater, soil moisture and runoff, is likely to change under a future climate. Hydrological models provide important insights into the translation of climate information to water partitioning at or below the land surface. However, drylands exhibit several key hydrological processes that are distinct from humid regions and which are typically omitted from most current hydrological models (Huang et al., 2017). The lack of simple, computationally efficient hydrological models for drylands undermines efforts to anticipate and plan for climatic and anthropogenic changes to water storage and fluxes in catchments, with implications for water resources for ecosystems and society (Huang et al., 2017). Drylands cover around 40 % of the global land surface (Cherlet et al., 2018) and support a population of around two billion people (White and Nackoney, 2003), yet there are no widely available, parsimonious models for simulating the dryland water balance. Climatically, dryland regions are characterised by high rates of evapotranspiration and low annual precipitation delivered with high spatial and temporal variability (Aryal et al., 2020; Wheater et al., 2007; Zoccatelli et al., 2019). Precipitation events are





characterised by high intensity and low duration rainfall over restricted spatial areas (Pilgrim et al., 1988). This results in a highly dynamic hydrological system prone to flash flooding, and also to water scarcity and food insecurity, societal risks that are exacerbated by climate change, population growth and dryland expansion (Cuthbert et al., 2019a; Giordano, 2009; Huang et al., 2017, 2015; Reynolds et al., 2007; Siebert et al., 2010; Taylor et al., 2012; Wang et al., 2017).

In drylands, runoff occurs mainly as infiltration excess (Hortonian) overland flow due to high intense precipitation events, and it leads to the development of short-lived streamflow in ephemeral streams. These ephemeral streams play an important role in the water balance because high transmission losses of water through porous stream beds are the main source of aquifer recharge in such environments - a mechanism called focused recharge (Abdulrazzak, 1995; Coes and Pool, 2007; Cuthbert et al., 2016; Goodrich et al., 2018, 2013; Schreiner-McGraw et al., 2019; Shanafield et al., 2014). In contrast, diffuse recharge, which is the result of local infiltration of water below the evaporation zone within the soil, is typically limited in drylands due to low precipitation and high rates of evapotranspiration (Schreiner-McGraw et al., 2019; Taylor et al., 2013). These conditions result in dryland environments having no significant long-term storage of water within soils (Huang et al., 2017; Pilgrim et al., 1988).

The complexity of rainfall regimes, runoff generation processes and subsurface flow paths in drylands create challenges for data collection, resulting in a paucity of data, and consequently it restricts the use of numerical models to enhance understanding of the water balance (Abbott et al., 1986; Cuthbert et al., 2019b; Ewen et al., 2000; Ivanov et al., 2004; Michaelides and Wainwright, 2002; Noorduijn et al., 2014; Schreiner-McGraw et al., 2019; Šimůnek et al., 2006; Wheater et al., 2007; Woodward and Dawson, 2000; Woolhiser, 1989). Existing hydrological models, operating at catchment to regional scales, are challenged in drylands due to their inherent assumptions about key flow processes and due to hard-coded parameterizations (e.g. physically-based models (Kampf and Burges, 2007)). Models also generally lack the ability to represent the development of ephemeral streams and their potential hydraulic interactions with groundwater systems (Quichimbo et al., 2020; Zimmer et al., 2020). Despite the recent improvement in models to include transmission losses (e.g. Hughes et al. (2006); Hughes (2019); Lahmers et al. (2019); Mudd (2006)), availability of appropriate numerical tools that allow a better description of surface-groundwater interactions are still limited at catchment, regional and global scales. Ephemeral flow in streams and losses to the subsurface are currently underrepresented in medium to large scale models, despite representing half of the global stream network length (Datry et al., 2017). Additionally, the degree of complexity of existing models and their inherently high computational cost does not allow for comprehensive sensitivity and uncertainty analysis, which would support the evaluation and interpretation of model results.

In this context, it is important for models to capture the linkages between the spatially and temporally variable climate, nonuniform runoff generation, soil moisture and focused groundwater recharge to support predictive capability of how the dryland water balance may shift with changes in climate. Models also need to include groundwater processes in drylands where the low regional hydraulic gradient governs the redistribution of groundwater resources, such as water availability for evapotranspiration in riparian areas (Maxwell and Condon, 2016; Mayes et al., 2020). Only a few large-scale hydrological models include gradient-based (diffuse) groundwater flow processes (de Graaf et al., 2015; Reinecke et al., 2019; Vergnes et al., 2012). These processes should be kept simple to make them transferable to different catchments regardless of their scale. Useful dryland models should also be able to employ the limited information available, while being numerically efficient enough to allow for evaluation of the model performance and uncertainty.

Here we present the development of a parsimonious model which considers the main processes that control the water partitioning, fluxes, and changes in water storage in dryland regions for estimation of runoff, soil moisture, actual evapotranspiration and groundwater recharge. We do not intend for this model to accurately simulate event-based flood hydrographs, for example, for flood hazard analysis. Instead, we aimed to develop a model that captures the long-term behaviour of the water balance in dryland regions. Here we apply and test our new model in the Walnut Gulch Experimental Watershed (WGEW), in SE Arizona, USA, where availability of high resolution data enabled us to evaluate the model performance.





## 2. DRYP: a parsimonious model for DRYland regions water Partitioning

**2.1. Model overview**

The main hydrological processes that control fluxes and storage of water in dryland regions are shown in Fig. 1a. The movement of water through the different storage components within the catchment is characterised as follows: spatially distributed rainfall falling during individual events over the surface is partitioned into *infiltration* and
*runoff*, depending on the temporal and spatial characteristics of the rainfall and the antecedent soil moisture conditions at the beginning of the rainfall event (Goodrich et al., 1997; Zoccatelli et al., 2019). Water infiltrated into the soil can be extracted by plant evapotranspiration and/or soil evaporation, or it can percolate to the water table as *diffuse recharge*. Runoff is routed to the nearest stream based on topographic gradient. In each stream reach, water may be added through groundwater discharge as *baseflow* or water may be lost through the porous
boundaries by *transmission losses* as it moves downstream. The volumes of both baseflow and transmission losses are dependent on the water table depth (Quichimbo et al., 2020). Transmission losses into the near-channel alluvial sediments increase the water available for plant evapotranspiration in the *riparian zone* and also generate *focused recharge* when the water holding capacity of the sediments in the riparian zone is exceeded (a la Schreiner-McGraw et al., 2019). Groundwater discharge into streams depends on the hydraulic gradient, occuring when the
water table elevation is higher than streambed elevation. Additionally, when the water table is close to the surface, *capillary rise* increases the root zone water availability for riparian plant evapotranspiration. Finally, anthropogenic activities, such as localized stream and groundwater abstraction as well as irrigation, may affect the storage and fluxes of the water balance.

The only forcing variables in DRYP are spatially explicit fields of precipitation and potential evapotranspiration. The partitioning of the water balance then depends on the combination of this forcing and its interactions with spatially distributed parameters representing topography, land cover, soil hydraulic properties, hydrogeological characteristics of the aquifer, and anthropogenic activities (Fig. 1b). Hydrological processes in DRYP are structured into three main components: i) a surface water component (SW) where precipitation is partitioned into
infiltration and overland flow, which is then routed through the model domain based on the topographic gradient; ii) an unsaturated zone (UZ) component that represents the soil and a riparian area parallel to streams; and iii) an saturated zone (SZ) component which represents groundwater flow (Fig. 1c). All three components in DRYP are discretized as square grid cells, and all components are vertically integrated into a computational one-way sequential scheme (Fig. 1c). However, all components are hydraulically interconnected, allowing for gradient-
driven, and potentially bi-directional water exchange (Fig 1c and d).

Figure(1)

DRYP is written in Python and uses the Python-based Landlab package, which has versatility to handle grided
datasets and model domains (Barnhart et al., 2020; Hobley et al., 2017a). DRYP is structured in a modular way to allow user flexibility to control the desired level of process and parameter complexity, as well as the grid size and time-stepping choices appropriate for the desired application of the model. The grid size is the same for all layers, but the time step for different components may vary flexibly as described below. All grid cells potentially consist of all the process elements shown in Fig. 1d. However, the stream and riparian components can be excluded if
stream channel characteristics are not provided, in which case all generated runoff in a cell will simply be routed to the next downstream cell with no additional losses or interactions.

For all cells, at the beginning of every time step, the input rainfall (P) is partitioned into surface runoff (RO) and infiltration (I) depending on the available water content of the unsaturated zone (UZ). Water in the UZ can be
extracted as actual evapotranspiration (AET), a combination of soil evaporation and plant transpiration, and/or percolate (R) to the saturated zone (SZ), depending on the water content and hydraulic properties of the unsaturated zone. If a cell is defined as a stream, transmission losses (TL) or groundwater discharge contributing to base flow (BF) and a riparian unsaturated zone (RUZ) are included in the local partitioning. The riparian zone is defined as an area parallel to the stream with a specified width. The riparian zone receives contributions from TL and a





volume of infiltrated water proportional to the riparian area. Water within the riparian zone can either percolate, becoming focused recharge or it can be extracted by plants as riparian evapotranspiration. Focused and diffuse recharge are combined as the main inputs to the SZ, which may also interact with the UZ depending on the water table elevation as it rises and falls through the simulation. The movement of water in SZ is driven by the lateral hydraulic gradient. Additionally, anthropogenic interactions in the model are implemented as localized fluxes

from the saturated zone (ASZ) and streams (AOF), whereas water abstraction for irrigation (AUZ) is delivered to the surface where it then contributes to infiltration into the unsaturated zone.

### 2.2 Surface Component

Two main processes are considered in the surface component: i) the partitioning of precipitation into infiltration and runoff, and ii) runoff routing and the partitioning of runoff into streamflow and transmission losses in

stream cells. These are described below.

### 2.2.1 Infiltration and runoff

The partitioning of precipitation into infiltration and runoff at the land surface is a key process in drylands and a potentially major source of uncertainty in the overall water partitioning for these regions. Hence, four different

infiltration approaches have been included in DRYP, which can be toggled on or off within the main control file (prior to simulation) to allow the user to experiment with different infiltration model structures. These approaches include two point-scale methods: the Philip infiltration approach and the Modified Green & Ampt method; and two upscaled methods for summarising infiltration over larger areas: the Upscaled Green & Ampt and the Multiscale Schaake approach.


*Method 1: Infiltration based on Philip's equation*
In this option, infiltration, $f$ [L T$^{-1}$] during a rainfall event is based on the explicit solution of the infiltrability depth of Philip's equation (Philip, 1957).

$$f(t_c) = \frac{1}{2}S_p t_c^{-\frac{1}{2}} + K_{sat},\qquad(1)$$

where: $K_{sat}$ is saturated hydraulic conductivity [L T$^{-1}$], $S_p$ is sorptivity [L$^2$ T$^{1/2}$], and $t_c$ is time since the beginning of the precipitation event [T]. The sorptivity term is estimated by using the following equation (Rawls et al., 1982):

$$S_p = \left[2K_s(\theta_{sat} - \theta)|\psi_f|\right]^{\frac{1}{2}},\qquad(2)$$

where: $\theta$ is volumetric water content [L$^3$ L$^{-3}$], $\theta_{sat}$ is volumetric water content at saturated conditions [L$^3$ L$^{-3}$], and $\psi_f$ is suction head [L] estimated as (Clapp and Hornberger, 1978):

$$|\psi_f| = \psi_a \frac{2\lambda+2.5}{\lambda+2.5},\qquad(3)$$

where: $\psi_a$ is maximum suction head [L], and $\lambda$ is a parameter that represents the pore size distribution of the soil [-] (Clapp and Hornberger, 1978).


The total infiltration depth in any given cell, $I$ [L], during a precipitation event is estimated by solving the integral of Eq. (1) over the event duration. The integral of Eq. (1) is solved using the time compression approach (TCA) (Holtan, 1945; Mein and Larson, 1973; Sherman, 1943; Sivapalan and Milly, 1989), assuming that infiltration after ponding depends on the cumulative infiltrated volume. Therefore, to match the initial infiltration rate at the

beginning of each time step with the infiltration at the end of the previous time step, the start time of infiltration





is shifted to match the total cumulative infiltration. A more detailed description and the analytical solution of the approach can be found in Assouline (2013) and Chow et al. (1988).

*Method 2: Infiltration based on a Modified Green - Ampt method*

We have implemented a modified version of Green & Ampt approach defined by the following equation (Michaelides and Wilson, 2007; Scoging and Thornes, 1979):

$$f(t_c) = K_{sat} + \frac{B}{t_c},$$  (4)

where: $B$ represents initial suction head [L], $t_c$ is the same as Eq. (1); here we use sorptivity (Eq. 2) as a proxy of the initial head owing to the nonlinear dependency of sorptivity on the water content of the soil.

The integral of Eq. (4) was also solved using the time compression approach (Holtan, 1945; Mein and Larson, 1973; Sherman, 1943; Sivapalan and Milly, 1989). However, since there is no explicit solution for Eq. (4), we 215     used an implicit solution.

*Method 3: Infiltration based on an Upscaled Green - Ampt method*

This method is based on the semi-analytical solution of the Green and Ampt equation for spatially heterogeneous hydraulic conductivity developed by Craig et al. (2010):


$$\bar{I}(t_c) = \frac{p}{2}\text{erfc}\left(\frac{ln\,(pX)-\mu_Y}{\sigma_{Y\sqrt{2}}}\right) + \frac{1}{2X}\log|K_{sat}|\text{erfc}\left(\frac{\sigma_Y}{\sqrt{2}} - \frac{ln\,(pX)-\mu_Y}{\sigma_{Y\sqrt{2}}}\right) + p\int_0^{X(t_c)} \varepsilon(X(t),K_{sat})\;f_k(K_{sat})dK_{sat}$$  (5)

where: $\hat{I}$ is the mean infiltration rate [L T$^{-1}$], $p$ is the precipitation rate [L T$^{-1}$], $t_c$ the same as Eq. (1), $f_k$ is the probability density function of $K_{sat}$, $\mu_Y$ and $\sigma_Y$ are mean and standard deviation of the log saturated hydraulic 225     conductivity, $\mu_Y = \ln|K_{sat}| - \frac{1}{2}\sigma_Y$, $X$ is a dimensionless time estimated as:

$$X = \frac{1}{1+\frac{\alpha}{Pt_c}},$$  (6)

where: $\alpha = |\psi_f|(\theta_{sat} - \theta)$, with $\psi_f$ representing the suction head.


The $\varepsilon(X,K_s)$ in Eq. (5) is an error function that can be estimated by the following approximation (Craig et al., 2010):

$$\varepsilon \approx 0.3632 \cdot (1-X)^{0.484} \cdot \left(1 - \frac{K_{sat}}{pX}\right)^{1.74} \left(\frac{K_{sat}}{pX}\right)^{0.38}$$  (7)


The $f_k(K_s)$ is assumed as a lognormal distribution following Craig et al. (2010):

$$f_K(K_{sat}) = \frac{1}{K_{sat}\sigma_Y\sqrt{2\pi}}\exp\left(-\frac{(\ln(K_{sat})-\mu_Y)^2}{2\sigma_Y^2}\right)$$  (8)

As suggested by Craig et al. (2010), we solve the integral of the Eq. (5) efficiently using a 2-point Gauss-Lagrange numerical integration method.

*Method 4: Infiltration based on the Multi-scale Schaake method*

The Schaake et al. (1996) approach is based on the assumption that rainfall and infiltration rates follow an 245     exponential distribution to approximate spatial heterogeneity of soil properties. Therefore, the spatially averaged infiltration $I$ [L] is estimated as:





$$I = \frac{PI_c}{P+I_c}, \tag{9}$$

where: $P$ is total rainfall [L] and $I_c$ is cumulative infiltration capacity [L].

Infiltration capacity is estimated as (Schaake et al., 1996):

$$I_c = (\theta_{sat} - \theta)\big(1 - exp(-k_{dt})\big), \tag{10}$$


where $k_{dt}$ is a constant that depends on soil hydraulic properties.

Following Chen & Dudhia (2001) we define $k_{dt}$ as:

$$k_{dt} = k_{dt_{ref}} \frac{K_{sat}}{K_{ref}}, \tag{11}$$

where: $K_{ref}$ [L T$^{-1}$] is a reference hydraulic saturated conductivity equal to $2 \times 10^{-6}$ m s$^{-1}$ (Chen and Dudhia, 2001; Wood et al., 1998) and the parameter, $K_{dtref}$, is specified as a scale calibration parameter.

**2.2.2 Runoff routing and transmission losses**

Rainfall that does not infiltrate (i.e. precipitation, $P$, minus infiltration, $I$) into the unsaturated component is routed over the model domain based on topography. The flow routing scheme varies depending on whether a cell is defined as a stream. A simple flow accumulation approach is used in cells without a defined stream, whereas for defined stream cells, an additional flux term is added to the flow accumulation approach to account
for groundwater interactions via the riparian zone. This flux will either be a transmission loss or a baseflow contribution from the saturated component.

*Flow routing in cells without streams*
Runoff produced in any given cell is instantaneously routed to the next downstream cells using the flow accumulation approach implemented in Landlab (Braun and Willett, 2013; Hobley et al., 2017b).
The next downstream cell is estimated using a D8 flow direction approach (8 potential directions based on adjacent cells). The flow accumulation method adds the amount of runoff from the upstream cells:

$$Q_i = \sum_{i=1}^{N} Q_{in_i}, \tag{12}$$

where: $Q_{in}$ [L$^3$] is the volume of water that discharges from upstream cells into the current cell $i$, $N$ is the number of upstream cells discharging into the current cell. and $Q_i$ [L$^3$] is the volume of water in the cell.

*Flow routing in stream cells*
In defined stream cells, the amount of water entering the cell, $q_{in}$ [L$^3$ T$^{-1}$], is instantly reduced by any transmission
losses, $i_{ch}$ [L$^3$ T$^{-1}$], and any remaining water, $q_{out}$ [L$^3$ T$^{-1}$], is moved to the next downstream cell:

$$q_{out} = q_{in} - i_{ch} \tag{13}$$

Water from the upstream cell, $q_{in}$, is assumed to be released to the next cell following a linear reservoir approach:
$$q_{in} = q_0 e^{-k_T t^*}, \tag{14}$$

where: $k_T$ [T$^{-1}$] is a recession term that is equal to the inverse of the residence time of the streamflow at each cell, $t^*$ represents time [T], and $q_0$ is the initial flow rate of water entering the channel, estimated as:
$$q_0 = (Q_{in} + S_{SW} - Q_{ASW})k_T, \tag{15}$$





where: $Q_{ASW}$ [L$^3$] is the volume of water abstracted from the stream, and $S_{SW}$ [L$^3$] is water stored in the channel.

It is assumed that the sediments in the streambed are homogenous. Consequently, the rate of infiltration depends on the wetted perimeter of the channel, and the infiltration rate, $i_{ch}$, at the stream cell is estimated assuming a unit gradient Darcian flow across the wetter perimeter:


$$i_{ch} = K_{ch}(2y + W)L_{ch} \qquad (16)$$

where $K_{ch}$ [L T$^{-1}$] is saturated hydraulic conductivity of the streambed, $L_{ch}$ [L] is channel length for a given cell, $W$ is channel width [L], and $y$ is streamflow stage [L]. If the rate of water entering the stream cell is less than the potential channel infiltration rate, flow to the next downstream cell is set to zero (all water is lost via infiltration) and $i_{ch} = q_{in}$.


Stream stage, $y$, is estimated by assuming that flow velocity does not change along the channel in any given cell (no flow acceleration). Therefore, the streamflow stage and the volume at any time along the channel are kept constant in any given stream cell. A constant velocity approach assumes that there are no backward effects on the streamflow routing approach. Thus, the stream stage is estimated as the height of the rectangular prism with area

$A = W\,L_{ch}$ and volume at time $t$ as:

$$y = \frac{q_{in}}{A} \qquad (17)$$

After substituting Eq. (17) into Eq. (16) and then into Eq. (13), the time integral of Eq. (13) represents the total

amount of water, $Q_{out}$ [L$^3$], that moves to the next downstream channel cell (becoming $Q_{in}$):

$$Q_{out} = \int_0^{\min[t_{q=0}, \Delta t]} \left[ q_0 e^{-k_T t} - L_{ch} K_{ch} \left( 2 \frac{q_0 e^{-k_T t}}{W L_{ch}} + W \right) \right] dt \qquad (18)$$

Note that the time step choice is important to bear in mind with respect to the size of the catchment modelled,

since it represents the minimum travel time for flow to reach the catchment outlet.

The amount of water stored in the channel is estimated by applying a mass balance of all inputs and outputs of the channel:


$$S_{RO}^t = Q_{in} + S_{SW}^{t-1} - Q_{ASW} - Q_{TL} - Q_{out} \qquad (19)$$

where: $t$ represents the current time step, and $Q_{TL}$ [L$^3$] is transmission losses estimated as the integral of the second term of Eq. (18). The total of $Q_{TL}$ is restricted to the storage available in the aquifer:


$$Q_{TL} = \min[Q_{TL}, \max[(z - h)\,A\,S_y, 0] \qquad (20)$$

where: $z$ is the surface elevation [L], $h$ is water table elevation [L], $A$ is the area of cell [L$^2$], and $S_y$ is aquifer specific yield [-].

### 2.3 Unsaturated Component

Water infiltrated into the soil or through the stream channel becomes a flux input to the UZ (Fig. 1d). The unsaturated component comprises the soil and the riparian zone, both of which are simulated using a linear 'bucket' soil moisture balance model (Fig. 2a), following an approach similar to the FAO water balance model (Allen et al., 1998):


$$\Delta S_{UZ} = I + Q_{TL} - AET - R, \qquad (21)$$



where: $\Delta S$ represents storage change [L], $AET$ represents actual evapotranspiration rate [L T$^{-1}$] and $R$ represents potential recharge rate [L T$^{-1}$]. The term $Q_{TL}$ is only defined for stream cells. Diffuse potential recharge results from the local vertical percolation of the unsaturated zone, whereas focused potential recharge is produced in the riparian unsaturated zone (see Fig. 1).


Figure(2)

The amount of water available for plant evapotranspiration in the UZ, $L$ [L], is estimated as the product of the rooting depth, $D_{root}$ [L], and the relative water content, $\theta$ [L$^3$ L$^{-3}$]. The maximum amount of water that the soil can store is limited by the field capacity of the soil ($L_{fc}$), whereas the minimum amount is constrained by the wilting point ($L_{wp}$). Thus, the total available water, $L_{TAW}$, for plant transpiration is estimated by the difference between $L_{fc}$ and $L_{wp}$ (see Fig. 2).


The potential amount of water that plants can remove water from the UZ as transpiration, $PET$ [L T$^{-1}$], which is the result of the product between a crop coefficient, $k$ [-], and the reference potential evapotranspiration, $ET_0$ [L T$^{-1}$] (Allen et al., 1998). When there is enough water to supply plant energy demands, water can be extracted from the UZ at a rate equal to $PET$. However, when there is not enough water in the UZ to supply the $PET$, plants are considered to be under stressed conditions and the actual evapotranspiration ($AET$) is constrained as:



$$AET = I + \beta(PET - I) \tag{22}$$

where: $\beta$ is a dimensionless parameter that depends on the water content and is estimated by:


$$\beta = \frac{L - L_{TAW}}{L_{TAW}(1-c)} \tag{23}$$

where: $c$ is the fraction of $L_{TAW}$ [-] at which plants can extract water from the UZ without suffering water stress, and set to 0.5 as recommended by the FAO guidelines (Allen et al., 1998) , although this can be varied in DRYP.

If, after accounting for infiltration and $AET$, there is a surplus of water in the soil that exceeds the field capacity, diffuse recharge ($R$) to the groundwater system occurs. If the model is run at daily time steps, we assume that all water content above field capacity will percolate and produce $R$. However, for sub-daily time steps, it is more realistic to assume that the soil slowly releases water as $R$ when it is above the field capacity, depending on the soil water retention curve. Hence, in this case we assume that percolation to the water table depends on the water

content and occurs only under the influence of gravity as follows:

$$D_{UZ} \frac{d\theta}{dt} = -K(\theta) \tag{24}$$

where $K(\theta)$ is estimated by using the Brooks and Corey (1964) relations and Clapp and Hornberger (1978)

parameters (see Eq (3)):

$$K(\theta) = K_{sat} \left( \frac{\theta}{\theta_{sat}} \right)^{(2\lambda + 2.5)} \tag{25}$$

We then substitute Eq. (25) into Eq. (24) and assume that the soil drains immediately into the groundwater

component after evapotranspiration loss. Hence, an analytical solution based only on drainage without considering other inputs or outputs is specified by:

$$\theta = \exp\left(-(2\lambda - 1.5) \log \left| \theta^{-2\lambda - 1.5} - \frac{\Delta t(2\lambda + 1.5) K_{sat}}{D_{UZ} \theta_{sat}^{2\lambda + 2.5}} \right| \right) \tag{26}$$





The UZ model component in DRYP can also change its behaviour when the head in the SZ component beneath restricts downward movement of water. This case is described below in Section 2.4 (Unsaturated – Saturated zone interactions).

**2.4 Saturated Component**

Lateral saturated flow underneath the unsaturated zone assumes the Dupuit-Forchheimer conditions for the Boussinesq equation and Darcian conditions for flow in/out of each model cell:

$$\frac{\partial h}{\partial t} + q_s + q_{riv} = \frac{1}{S_y} \nabla \cdot (-K_{sat} h \nabla h) + R - Q_{ASZ} \tag{27}$$

where: $K_{aq}$ is the saturated hydraulic conductivity of the aquifer [L T$^{-1}$], $S_y$ is the specific yield [-], $q_s$ is saturation excess [L T$^{-1}$] (see Sect. 2.4.1), $q_{riv}$ is discharge into stream [L T$^{-1}$] (see Sect. 2.4.2), $Q_{ASZ}$ [L T$^{-1}$] is any groundwater abstraction, $\nabla$ represents the gradient operator and $\nabla \cdot$ represents the divergence operator. Where the saturated thickness of the aquifer is relatively constant over the simulation period, transmissivity, $T$ [L$^2$ T$^{-1}$], (the product of the aquifer thickness and the saturated hydraulic conductivity of the aquifer), may be held constant, hence 410 linearising Eq. (25). Additionally, an exponential function based on Fan et al. (2013) has been added to represent the reduction of transmissivity in relation to depth:

$$T = K_{sat} f_D \, exp\left(-\frac{z-h}{f_D}\right), \tag{28}$$

where: $f_D$ is effective aquifer depth [L]. These different transmissivity parameterisation options can be toggled on or off in the main model control file.

Equation (27) is solved using a forward time central space (FTCS) finite difference approach. FTCS is an explicit finite difference approximation whose solution is sensitive to grid size and time step. Thus, in order to obtain a 420 stable convergence of Eq. (27), a time variable approach was adopted. The maximum allowable time step for the saturated component is estimated based on the Courant number criteria (we use 0.25 as a default value but this may be changed by the user):

$$\frac{T\Delta t}{S_y \Delta x^2} \leq 0.25 \tag{29}$$


If the maximum time step of the SZ component is greater than the time step of the minimum time step of the any other component of the model, the time step of the SZ component is reduced to the time step of the minimum time step of the model (see Sect. 2.6 for more details of the model time step options).

**2.4.1 Unsaturated - Saturated zone interactions**


Unsaturated - saturated zone interactions are implemented using a variable depth unsaturated zone as follows (Fig. 3a). Unsaturated zone thickness ($D_{uz}$) is equal to the rooting depth when the water table elevation ($h$) is below the rooting depth, but when the water table is above the rooting depth the thickness of the unsaturated zone is reduced to the depth of the water table:


$$D_{uz} = min[D_{root}, z - h] \tag{30}$$

When the water table is below the rooting elevation, $z_{root}$, there is no two-way interaction between the soil and the groundwater compartment (only one-way, as recharge), so no updates to the water table elevation are required 440 (see Fig. 3a, left panel). However, when the water table crosses the $z_{root}$ threshold, either via recharge or lateral groundwater flow, the water table is updated depending on the change in groundwater storage:



$$\frac{\Delta S_{SZ}}{\Delta t} = \nabla \cdot (-K_{sat} h \nabla h) + R - Q_{ASZ} \tag{31}$$

where: $\Delta S_{SZ}$ is the change in groundwater storage per unit area [L$^3$ L$^{-2}$]. Specifically, if an SZ cell is being
recharged and the water table rises past the rooting depth in a given timestep, the water table is updated according
to:

$$h_t = \frac{1}{\theta_{sat} - \theta_t} \left[ \Delta S_{SZ} - (z_{uz} - h_{t-1}) S_y \right] + z_{uz} \tag{32}$$

whereas, when the water table is draining and passes the rooting depth in a given timestep:

$$h_t = -\frac{1}{S_y} \left[ \Delta S_{SZ} - (h_{t-1} - z_{root})(\theta_{sat} - \theta_{fc}) \right] + z_{root} \tag{33}$$

When the water table is above the rooting depth elevation, the water table elevation will be updated according to:

$$h_t = \frac{\Delta S_{SZ}}{\theta_{sat} - \theta_{fc}} + h_{t-1} \tag{34}$$

while if it is below the rooting depth elevation, the water table elevation is simply:

$$h_t = \frac{\Delta S_{SZ}}{S_y} + h_{t-1} \tag{35}$$


When the water table is above $z_{root}$, there is more water potentially available for evapotranspiration, since it can
be taken from the groundwater reservoir via capillary rise or direct root water uptake. Thus, the potential maximum
amount of water taken up from the groundwater reservoir, $PAET_{SZ}$ [L T$^{-1}$], is computed as the remaining $PET$ after
$AET$ from the unsaturated component as:


$$PAET_{SZ} = PET - AET \tag{36}$$

For a shallow water table, upward capillary fluxes may also be taken from the groundwater reservoir. The rate of
actual evapotranspiration from the SZ ($AET_{SZ}$), including both plant water uptake and capillary rise, is thus
estimated as a linear function of the water table depth as follows:

$$AET_{SZ} = \max \left[ PAET_{SZ} \left( \frac{h - z_{root}}{D_{root}} \right) \Delta t, 0 \right] \tag{37}$$

### 2.4.2 Surface–groundwater interactions

Surface - groundwater interactions are characterised in DRYP through transmission losses as described in Sect.
2.2.2. In addition, when the water table intersects a cell's defined streambed elevation it produces discharge into
the stream, $q_{riv}$ [L T$^{-1}$], and when the water table reaches the ground surface it produces saturation excess, $q_s$ [L
T$^{-1}$] (Fig. 3b) (Eq. (27)).

Discharge into streams, $q_{riv}$, is quantified using a head-dependent flux boundary condition (similar to that used in
MODFLOW (Harbaugh, 2005)) as:

$$q_{riv} = C(h - h_{riv}) \tag{38}$$

where: $C$ is a conductance term [L$^2$T$^{-1}$] estimated as:

$$C = \frac{K_{ch} L_{ch} W}{0.25 \Delta x} \tag{39}$$



To avoid numerical instabilities, we use a regularisation approach implemented via a smooth switch between the flux boundary condition and a constant head boundary (and vice versa) using a convex function (Marçais et al., 2017):

$$q_s = f_u\left(\frac{h-h_b}{z-hb}\right)f_g(\nabla \cdot (-K_{sat}h\nabla h) + R - q_{riv}) \tag{40}$$

where: $h_b$ is aquifer bottom elevation [L], $f_u$ is the continuous function between [0,1] specified as (Marçais et al., 2017):

$$f_u = \exp\left(-\frac{1-u}{r}\right) \tag{41}$$

where $r$ is a dimensionless regularisation factor $r > 0$, which has been specified as 0.001 following Marçais et al. (2017). $f_g$ is the Heaviside step function.

$$f_g = \begin{cases} 0, u < 0 \\ u, u \geq 0 \end{cases} \tag{42}$$

After both $q_s$ and $q_{riv}$ are estimated, their corresponding volumes are estimated by multiplying the flow rate, the timestep and the corresponding surface area (cell or stream). The volume is then added as additional runoff in the surface component (Sect. 2.2.2). The water table is updated to its topographical elevation and kept as a constant head boundary condition. The boundary switches back to a flux condition if the water table drops back below the water table.

Figure(3)

### 2.5 Numerical implementation and time step

DRYP is a fully open-source, grid-based model with a layer-based structure, developed using the Landlab architecture (Hobley et al., 2017a) and its Python library. Landlab was chosen due to the versatility and its modular design that allows the user to plug in multiple modules for different levels of complexity and processes using grid-based objects (Barnhart et al., 2020; Hobley et al., 2017a).

Since most hydrological processes in DRYP, except the SZ component and the modified Green & Ampt infiltration, are described according to explicit-analytical solutions, it possible to run DRYP at hourly or sub-hourly time steps at a low computational cost.

The three main DRYP components (i.e. surface, unsaturated and saturated components), can run at different time steps, from sub-hourly to daily. The riparian zone of the unsaturated component can be also run at a different time step to that of the unsaturated component. Where different time steps are used between components, the fluxes and state variables are temporally aggregated in DRYP by accumulating and/or averaging them over the specified time step as appropriate and then transferring them to the next component. In addition, and as described above, for the saturated component, an internal time step is also automatically considered to ensure the stability of the numerical solution.

### 2.6 Model input files and parameter settings

DRYP requires spatial characterisation of key input parameters and data including a digital elevation model (DEM), channel properties in cells where streams are explicitly defined (length, width and saturated hydraulic conductivity), land cover (plant rooting depth), various soil hydraulic properties, and aquifer properties (specific yield, aquifer thickness, and saturated hydraulic conductivity) (Fig. 1). A summary of model parameters for the





different model components and structures is presented in Table 1. If parameters are not provided, 'global' default
values are used as defined in Table 1.


Precipitation and potential evapotranspiration are the only forcing variables and can be supplied as either spatially
variable gridded data sets in netCDF format or as spatially uniform values for each time step. Gridded data sets
must be interpolated or aggregated to match the model grid resolution.

Table(1)

### 3.    Model Evaluation Methods

### 3.1 Evaluation using synthetic experiments


The use of synthetic experiments is an important aspect of model development in hydrology which is welcome
but not used often (Clark et al., 2015). The objective of synthetic experiments is to better understand the structural
controls on the physical processes represented in the model, for example, on groundwater-soil interactions (Batelis
et al., 2020; Rahman et al., 2019). Here we perform a set of numerical experiments to evaluate the stability and
convergence of DRYP components, particularly the coupling of both the surface and unsaturated zone with the
groundwater component. Convergence and stability of the numerical solution of the groundwater component using
the FTCS finite difference approach and the regularization have been well documented in different studies (e.g.
Anderson et al., (2015); Marçais et al., (2017); Wang and Anderson, (1982)). Hence here we provide only a
qualitative evaluation of the model performance with respect to the desired skill of the model to seamlessly allow
interactions between groundwater and the land surface and surface water components.

The geometry of the model domain for these tests consisted of a tilted-V catchment (Fig. 4) with a size of 7×10
square cells on a 1-km resolution grid. Land use and soil hydraulic characteristics were specified as uniform over
the entire model domain, and the saturated zone was considered as a homogenous and unconfined aquifer.
Boundary conditions were specified as no-flow boundaries for all sides as well as at the bottom of the model
domain. The initial water table was set as a horizontal plane at the level of the catchment outlet (100 m) for all
simulations (Fig. 4). For experimental purposes, hydraulic characteristics of both the unsaturated and saturated
zone were arbitrarily chosen. Thus, a loamy sand soil texture with $K_{sat}$ = 29.9 cm h$^{-1}$, $\theta_{sat}$ = 0.40, $\theta_{fc}$ = 0.175, and
$\theta_{wp}$ = 0.075 was chosen for the unsaturated zone, whereas, for the saturated zone, the hydraulic conductivity of the
aquifer ($K_{aq}$) was specified as 6 m d$^{-1}$ and the specific yield ($S_y$) was set as 0.01. The high value of $K_{aq}$ combined
with $S_y$ and boundary conditions of the aquifer were applied in order to allow a fast increase/decrease of the water
table and the observation of surface-groundwater interaction in a short period of time.

Three main scenarios were analysed by using synthetic time series of precipitation and evapotranspiration and
changing hydraulic parameters of the UZ as follows:
1.   An 'Infiltration - discharge' scenario, where all precipitation was allowed to infiltrate into the
catchment and no infiltration excess was produced over the model domain.
2.   An 'Infiltration-evapotranspiration-discharge' scenario was simulated by adding a time variable
potential evapotranspiration as input into the model.
3.   An 'Infiltration-runoff-evapotranspiration-discharge' scenario was designed to evaluate the
production of runoff and focused groundwater recharge, as well as groundwater discharge. For this
last scenario, the saturated hydraulic conductivity of the soil was decreased by one order of
magnitude to produce infiltration excess and consequently, runoff.

For all three scenarios, precipitation events were specified at a constant value of 0.25 [mm h$^{-1}$] over 10 days
followed by a 20-day dry period. Potential evapotranspiration was specified as a sinusoidal function with a 24-
hour period and a maximum rate of 0.10 [mm h$^{-1}$]. These experimental values of precipitation and
evapotranspiration combined with the hydraulic properties of the unsaturated and saturated zone allowed a visual
evaluation of surface-groundwater interactions under different conditions, such as increasing and decreasing water
table through the model run and its interaction with the unsaturated zone.




Figure(4)

**3.2 Model evaluation based on observed catchment data at Walnut Gulch, USA**

In addition to evaluating the DRYP model with synthetic experiments, the model was also evaluated at the Walnut
Gulch Experimental Watershed (WGEW), a 149km$^2$ basin near Tombstone, Arizona, USA (31° 43'N, 110° 41'W)
(Fig. 5). The climate of the region is semi-arid with low annual rainfall, with a long-term average of 312 mm/yr
(Goodrich et al., 2008). The ephemeral channels of WGEW are comprised of mixed sedimentary beds sourced
that promote high transmission losses, leading to downstream declining discharge in all but the largest streamflow
events (Michaelides et al., 2018; Singer and Michaelides, 2014). WGEW was chosen because it has a long and
spatially explicit record of runoff (Stone et al., 2008) for multiple flumes as well as high density event-based
rainfall data for 95 operational gauging stations (Goodrich et al., 2008) which were used to analyse trends in
rainfall characteristics (Singer and Michaelides, 2017) and from which the STORM model was created (Singer et
al., 2018). In addition, water content from a cosmic-ray neutron sensor as well as latent heat flux from Eddy
covariance flux tower are also available in the basin (Emmerich and Verdugo, 2008; Zreda et al., 2012). Together
these data provide an ideal opportunity to assess many components of a model of dryland water balance and
partitioning (Emmerich and Verdugo, 2008; Goodrich et al., 2008; Keefer et al., 2008; Scott et al., 2015; Stone et
al., 2008). The water table at WGEW is deep (~50 m at the catchment outlet), so the potential interaction between
surface and groundwater is generally unidirectional (Quichimbo et al., 2020).


Figure(5)

*Model setting, inputs and parameters*

For WGEW, model simulations were performed using the modified Green and Ampt infiltration approach because
of its ability to describe the high potential infiltration rates at the beginning of the precipitation event, which is
particularly important in this setting. The time step of both the surface component and unsaturated component was
specified as one-hour, whereas we use a time step of one day for the riparian zone to reduce computational time.
The high temporal resolution for the unsaturated component was used to capture the observed high intensity, low
duration rainfall at WGEW, as well as the influence of diurnal fluctuations in evapotranspiration. Since the water
table is deep below the ground surface and surface water groundwater interactions are known to be limited, the
groundwater component was not included in model simulations for WGEW.

Spatial and temporal information required as inputs and for model parameters were obtained for WGEW from
https://www.tucson.ars.ag .gov/dap/. A model domain of 104 × 41 square cells on a 300-m resolution grid was
developed. A digital elevation model with a spatial resolution of 30 × 30 metres was obtained from SRTM 1 Arc-
Second Global map (available at https://earthexplorer.usgs.gov). The DEM was aggregated by averaging cells to
the 300-m grid size. Textural characteristics of soil and land cover, obtained as polygon files from
https://www.tucson.ars.ag .gov/dap/, were converted into model gridded inputs by considering the feature with
the biggest area as the raster value. Based on the soil texture, baseline hydraulic properties required for modelling
were obtained from Rawls et al. (1982) and Clapp & Hornberger (1978). Values of field capacity and wilting point
required to estimate $L_{TAW}$ (Fig. 2a) were obtained assuming a matric potential of -33 kPa and -1500 kPa following
the FAO guidelines (Walker, 1989).

Stream positions were estimated from the 30 × 30 m DEM. The routing network at the 30 × 30 m grid resolution
was specified by defining a minimal upstream drainage area threshold of 65ha, which corresponds to the medium
stream network resolution specified in Heilman et al. (2008). Stream cells were then aggregated to the model grid
size, 300 × 300 m, to obtain the stream length at any given cell. Point measurements of rainfall were obtained
from 95 rainfall stations well distributed within the (Goodrich et al., 2008). Rainfall data, at every location,
were temporally aggregated to 1-hour and then spatially interpolated using a Natural Neighbour algorithm to a 30
× 30 m grid size to preserve the high spatial and temporal variability of station located at distances smaller than

the model grid size. Finally, rainfall was spatially aggregated to the grid size of the model domain. Potential evapotranspiration (*PET*) was calculated using hourly data from ERA5-Land reanalysis (Hersbach et al., 2020) because this dataset enabled high temporal resolution (1-hour) and its potential to drive hydrological and land surface models (Albergel et al., 2018; Alfieri et al., 2020; Tarek et al., 2020). Data from ERA5 have a spatial resolution of ~9km at the equator. The Penman-Monteith approach was chosen to estimate hourly *PET* due to its high accuracy to produce evapotranspiration values under different climates and locations, and also because it is considered a standard method by the FAO (Food and Agriculture Organization of the United Nations) (Allen et al., 1998).

High-resolution temporal measurements of runoff at three flumes (F01, F02, and F06) along the main Walnut Gulch channel were used in the evaluation of runoff generation (Fig. 5). To evaluate modelled soil moisture, we used data from a cosmic-ray neutron sensor station from the COSMOS network (Zreda et al., 2012), located within the Kendall subcatchment of WGEW (Fig. 5). The raw data (publicly available at http://cosmos.hwr.arizona.edu/Probes/StationDat/010/index.php) were corrected for atmospheric pressure (Hidroinnova, 2013), atmospheric vapour pressure (Rosolem et al., 2013), above ground biomass and variation in background intensity using the standardized data processing Cosmic-Ray Sensor PYthon tool (Power et al., 2021) for the period between mid-2010 and 2018.

Finally, data from the Ameriflux site Kendall Grassland (US-Wkg; available at https://ameriflux.lbl.gov/sites/siteinfo/US-Wkg), were used for evaluation of simulated *AET* (Fig. 5). Uncertainty in flux tower data is mainly attributed to instrumental and random errors and it increases with flux magnitude (Richardson et al., 2006; Schmidt et al., 2012). Mean relative errors for AmeriFlux sites are around -5 % with deviations of ± 16 % (Schmidt et al., 2012). Historical records from mid-2006 to 2018 were available for model evaluation.

### *Model sensitivity analysis and calibration*

An initial trial-and-error calibration of the model was performed to explore the parameter sensitivities of DRYP and to reduce the a-priori parameter ranges used in the second step. This first trial-and-error calibration considered only the performance of the model to represent streamflow at the catchment outlet (flume F01). The calibration was performed by applying spatially constant multiplicative factors $kW$, $kK_{sat}$, $kD$, $kK_{ch}$, $kk_T$, to model parameters $W$, $K_{sat}$, $D$, $K_{ch}$ and $k_T$, respectively. These parameters were used because they control the storage and the water partitioning of components (surface and subsurface components) in the DRYP model for WGEW. Parameters $W$, $K_{ch}$ and $k_T$ were assumed to be uniform over the entire catchment due to the lack of spatial information, whereas the rest of parameters listed in Table 1 vary depending on their mapped spatial distribution. The initial manual calibration enabled a set of parameter ranges to be defined for a Monte Carlo experiment to analyse the multi-parameter uncertainty of the model results. Then, a set of 1000 realisations was implemented for the analysis with parameters randomly generated using a uniform distribution.

The Generalized Likelihood Uncertainty Estimation (GLUE) framework (Beven and Binley, 1992, 1992) was used as the uncertainty analysis framework. The GLUE framework considers that, owing to the uncertainty of the input data, model structure and limitations of boundary condition, there are multiple set of parameters that can produce acceptable simulations. To determine which simulations were considered as acceptable (i.e. behavioural), we used a combination of two different 'goodness of fit' indices: Nash–Sutcliffe Efficiency (NSE) (Nash and Sutcliffe, 1970), and per cent bias (PBIAS) defined as follows:

$$NSE = 1 - \frac{\sum_{i=1}^{n}(O_i - S_i)^2}{\sum_{i=1}^{n}(O_i - \bar{O})^2} \tag{43}$$

$$PBIAS(\%) = 100 \cdot \frac{\sum_{i=1}^{n} O_i - \sum_{i=1}^{n} S_i}{\sum_{i=1}^{n} O_i} \tag{44}$$

where: $O$ represents the observation, $\bar{O}$ is the arithmetic mean of observations, $S$ represents the model simulations, and *n* is the number of observations.


In order to define behavioural models, a set of thresholds was specified for the three indices. For streamflow,
values of NSE higher than 0.50 and PBIAS less than 20 % (i.e. less than 1 % of the total water budget of the study
area) were considered as acceptable simulations. For soil moisture and actual evapotranspiration, only values of
NSE greater than 0.5 were also set.

In order to combine these measures into a single performance metric, models which did not meet these conditions
were assigned a value of zero, whereas the indexes were linearly scaled between 0 and 1 for rest of models. Scaling
of *NSE* values was performed according to the following range: 0 for the minimum value ($NSE = 0.5$) and 1 for
the maximum value of *NSE* which is also 1. For *PBIAS*, absolute values were scaled by considering the maximum
value ($PBIAS = 20$ %) equal to 0 and the minimum ($PBIAS = 0$) equal to 1. The combined performance measure
was calculated as the product of all indexes considered in the analysis:


$$p_i = \prod_{k=1,2,6} NS_k^* \cdot PBIAS_k^* \tag{45}$$

where $p^*$ is the combined performance measure for the *i*-th parameter set, the $*$ signifies scaled values, and *k*
represents the variable considered in the analysis.


For soil moisture, a direct comparison between observation and simulations was not possible due to differences
between the representative soil depths of measurements and simulations. Modelled soil moisture represents the
water content of the entire soil column specified by the rooting depth, whereas the observed soil moisture
represents the water content over a depth-averaged value, which can be characterized by an effective soil depth
that depends on the soil moisture itself (Franz et al., 2012). A direct comparison would result in the
misrepresentation of high values of observed soil water content by the model due to the attenuation of peak values
over larger soil depths. This problem has been solved by using exponential models that need to be calibrated by
using measurements at different soil depths (e.g. Albergel et al., (2008); Wagner et al. (1999)). Therefore, to
enable model-data comparisons that capture the variation of both high and low values of soil moisture
observations, we scaled observed soil moisture by the following expression:

$$O^* = O^\alpha + S_{min} \tag{46}$$

where: $*$ refers to the scaled value, and $\alpha$ is estimated by:


$$\alpha = \frac{\log|S_{max} - S_{min}|}{\log|O_{max} - O_{min}|} \tag{47}$$

The period between 01/01/2007 and 01/01/2018 was the temporal domain for model simulations at WGEW, with
a warm-up period of one year prior to this period. This period matches the overlapping period of streamflow
observations and flux tower observations. Soil moisture was evaluated for a shorter period of available data
01/10/2010 and 01/01/2018. Additionally, modelled soil moisture for COSMOS site was obtained by spatial
averaging the 9 cells located around the COSMOS station to match the effective COSMOS footprint diameter
(~700m) (Desilets and Zreda, 2013).





## 5. Results

### 5.1 Evaluation of synthetic experiments

Figure 6 shows the temporal variation of fluxes and state variables for the three simulated scenarios at two evaluation points located along the channel, one at the catchment outlet and the second 4 km from the catchment outlet. These results are in turn described below:

1. 'Infiltration - discharge' scenario (blue lines): when the precipitation falls over the catchment (Fig. 6a) it immediately infiltrates into the unsaturated zone, increasing the water content of the soil. Since there are no losses due to evapotranspiration, the water content steadily increases until it reaches field capacity (Fig. 6b). At field capacity, given that the soil cannot hold any excess water, it starts to release water as diffuse recharge. The soil remains at field capacity for the rest of the simulation, allowing the water from the next rainfall event to move directly to the saturated zone producing recharge (Fig. 6c). Recharge produces an increase in groundwater storage and consequently increases the discharge at the outlet of the catchment (Fig. 6d). In the early precipitation events, the contribution of groundwater discharge is minimal. However, this contribution keeps increasing until a dynamic steady-state is eventually reached (by 14600 hours, not shown in Fig. 6). Discharge closely follows the temporal variation of the precipitation, due to the high transmissivity of the aquifer and the saturation of the soil; a sharp increase in discharge means that precipitation has become the main contributor to discharge changes because the water table is at the surface (Fig. 6d).

2. 'Infiltration-evapotranspiration-discharge' (green lines): the addition of evapotranspiration in this experiment produces a reduction in soil water content (Fig. 6b). Since precipitation is much higher than evapotranspiration, soil moisture quickly reaches a dynamic steady-state at the end of the second precipitation event. At cells located close to the catchment outlet, the rise of the water table to ground level reduces the thickness of the unsaturated zone to zero, as no water can be infiltrated, the soil water content is kept at its highest value during the precipitation event. After the precipitation event, the rate of evapotranspiration, which is greater than the rate of lateral groundwater inflow, gradually reduces the amount of water in the cell. However, since the storage of the SZ keeps increasing, the thickness of the UZ decreases and the rate of lateral groundwater flow becomes greater than the rate of evapotranspiration, it also results in quick changes in the water content of the soil (Fig. 6b). Recharge is also reduced and, as expected, it only occurs when the soil moisture reaches field capacity (Fig. 6c). Discharge is also reduced as a result of decreased aquifer recharge due to upward losses by *AET*. For cells close to saturation, the storage in the groundwater reservoir is affected by evapotranspiration losses (not observed in right panel due to the y-axis scale), which in turn results in daily fluctuations in discharge that are inverse of evapotranspiration fluctuations.

3. 'Infiltration-runoff-evapotranspiration-discharge' scenario (red lines): a reduced $K_{sat}$ results in the development of infiltration excess overland flow. The rate of infiltration at the beginning of the precipitation event is high enough to provide water for evapotranspiration without reducing the soil water storage (Fig. 6b), which explains the similarity in soil moisture behaviour with the second scenario. When cells start to produce runoff as a result of infiltration-excess, discharge also starts to rise. At stream cells with a deep water table, the increase in streamflow is the result of flow accumulation along the channel during the precipitation event (e.g. Fig. 6d, left panel at 6600 to 6700 hours). At cells where the water table interacts with the surface, groundwater discharge increases gradually the streamflow at the catchment outlet at much longer temporal scales (Fig. 6d). At the catchment outlet, streamflow is also affected by the fluctuation of the water table due to the daily variation of evapotranspiration losses (Fig. 6d).

Figure (6)



Figure 7 shows the cumulative volumes of different components of the water balance as well as the cumulative mass balance error of the model. Mass balance errors are low in comparison to the total amount of water entering the catchment, with values less than 0.12 % for the first case (only precipitation). For the other two cases where evapotranspiration is included, errors are less than 0.02 %. The higher error for the first case scenario is attributed to the concentration of flow at the catchment outlet, which leads to an increase in the number of cells discharging into the surface and the channel and the resulting minor numerical artifacts.


Figure (7)

Coupling of surface and groundwater processes often results in numerical instabilities and in convergence problems (Batelaan and Smedt, 2004; Marçais et al., 2017). However, the results of these synthetic experiments
illustrate DRYP's ability to produce realistic hydrological process behaviours by providing a stable solution for representing surface-groundwater interactions without producing numerical artifacts. DRYP is effective at handling the complex coupling and dynamic switching of different types of hydraulic boundary conditions, producing acceptable results with negligible mass balance errors.

## 5.2. Model performance at WGEW

### 5.2.1. Spatio-temporal visualisation of model process simulation at WGEW

The ability of the model to capture the dynamics of dryland hydrological processes is illustrated for WGEW in Figure 8. The best model (see following section) captures the emergence of ephemeral flow conditions for specific
storms, as well as the spatio-temporal changes in soil moisture. It can be seen how, for a given initial soil moisture condition, the production of runoff due to a rainfall event falling over only the central part of catchment results in the concentration of flow along the stream. As water moves downstream, the stream loses water due to transmission losses, which ultimately consumes almost all the available water by the time runoff reaches the catchment outlet (flume F01 in Figure 8c).


Figure(8)

### 5.2.1. Characterisation of the temporal variation in simulated variables

Calibration using the trial-and-error method, showed that streamflow showed particular sensitivity to the parameters $K_{sat}$, $D$, $K_{ch}$ and $k_T$. This informed a set of parameters ranges that were used in the Monte Carlo analysis as follows: for hydraulic conductivity at the channel, $kK_{ch}$ 0.10 and 0.30, for $kK_{sat}$ 0.20 and 0.50, for $kk_T$ 3-10, and $kD$ 0.80 and 1.20. This resulted in 21 behavioural models with values of $p$ above zero. The calibrated parameters for the best simulation were $kK_{ch}$ = 0.21, $kAWC$ = 1.02, $kK_{sat}$ = 0.30, and $kk_T$ = 7.7. A factor of $kk_T$ = 7.7 applied
to default value of $k_T$ (0.083) represents a flow velocity of 0.41 m s$^{-1}$ in the channel.

*Soil moisture*
The DRYP model demonstrates skill at capturing the dynamics of the soil moisture (Fig. 9a) with values of NSE around 0.69. Discrepancies in the magnitude of peak values are likely the result of scaling, so simulations are not
able to account for the variation of the effective measurement depth of COSMOS water content estimates (Franz et al., 2013, 2012). The effective COSMOS measurement depth is greater for low values of soil water content (around 33 cm), whereas, for higher values of water content the effective measurement depth is shallow (around 16 cm). However, discrepancies may also reflect the limited ability of the soil moisture model to represent high variations occurring at shallow depths of the soil layer, due to the use of a single store.


*Evapotranspiration*
The DRYP model also captures well (NSE ~0.7) the seasonality and the overall temporal variation in evapotranspiration, a dominant component of the water budget in drylands (Fig. 9b), although peak values are generally overpredicted after long periods of dry conditions. Nevertheless, discrepancies between flux tower data
and simulated *AET* up to 15% for one year have been reported for grassland vegetation in previous studies (Scott,



2010; Twine et al., 2000), and such errors are mainly attributed to the inherent uncertainty in rainfall and latent heat flux measurements (Scott, 2010).

*Streamflow*

DRYP is also able to reproduce the seasonality and the monthly production of runoff at the outlet of the catchment (F01, NSE ~0.9) (Fig. 9c), as well as at the two upstream flumes (F02, F06) considered in the analysis (NSE > 0.60) (Figs. 9d and 9e). However, monthly values at flumes F02 and F06 are overpredicted in 2012, perhaps reflecting the development of a crusting layer in previous dry years (e.g. 2009, 2011), a process not included in the model. On the other hand, low production of runoff during wet years (e.g. 2015) may be attributed to the

energy of high intensity rainfall events removing such a crusting layer from the top of the soil, which in turn results in the increase of infiltration rates (Becker et al., 2018). Additionally, the spatial aggregation of the DEM causes slight inaccuracies in the estimated contributing areas for different streams. This affects not only the volume but also the timing of streamflow events, which may result in over/under prediction of streamflow events and may ultimately affect the overall water budget.


Figure (9)

*Water balance*

Precipitation shows high annual variability for the evaluated period, with the lowest value of 200 mm y$^{-1}$ in 2011

up to 400 mm y$^{-1}$ in 2015 (Fig. 9a), which translates into variability in the annual water partitioning for WGEW. For the evaluation period, 01/01/2007 to 01/01/2018, water balance estimates from the best model show that ~92 % of the total precipitation infiltrates into the soil (Figure 10). However, almost all infiltrated water returns to the atmosphere as evapotranspiration losses, representing 89 % of the total precipitation. A small proportion, ~3 % of the total precipitation, remains in the soil, and this stored water corresponds mainly to wetter years of the

simulation period (2014, and 2015). Only a small percentage, less than 0.03 %, percolates as diffuse recharge contributing to groundwater storage. Water that does not infiltrate into the soil (8 % of the precipitation) is routed downstream. However, this amount of water is consistently reduced by transmission losses, representing ~7 % of the precipitation. Water entering the riparian zone via transmission losses is partitioned into evapotranspiration and focused recharge. Evapotranspiration consumes up to 60 % of these transmission losses representing ~4.5 %

of the total precipitation. This is broadly consistent with previous studies showing values of 20 mm y$^{-1}$ or 5.5 to 7 % of the total precipitation (Renard, 1970; Renard et al., 2008). The amount of surface water leaving the catchment represents less than 1.0 % of the total amount of precipitation falling over the catchment. These values highlight the impact of transmission losses on the streamflow and aquifer recharge. The main contributor to the total amount of groundwater recharge is focused recharge (~2.5 % of precipitation).


Figure(10)

## 6. Conclusions

We have developed and presented a parsimonious model to estimate water partitioning in dryland regions ("DRYP"). We have provided a technical description of all components of DRYP and evaluated it under different scenarios. We first evaluated the ability of DRYP to provide stable numerical simulations of the interaction of surface and subsurface components through synthetic model experiments. Then, we evaluated DRYP using streamflow, soil moisture, and evapotranspiration data from the semi-arid Walnut Gulch Experimental Watershed

(Arizona, USA). We tested the ability of the model to produce behavioural simulations based on multi-parameter Monte-Carlo experiments evaluated against a range of objective performance metrics. Numerical experiments over a synthetic model domain showed that DRYP shows skill at producing stable simulations for the main components of the water balance with low mass balance errors (< 0.12 %). Thus, DRYP shows the potential to be applied in environments where surface-subsurface interactions play an important role in the overall mass balance

of the catchment.



For Walnut Gulch, DRYP effectively captures the spatio-temporal variability of the main components of the dryland water balance at monthly time scales. We find that focused recharge represents ~2.5 % of the total amount of rainfall, whereas diffuse recharge is below 0.03 %. Evapotranspiration is the dominant process representing 90% of water leaving the catchment. Evapotranspiration from riparian areas also plays an important role in groundwater recharge since the amount of water becoming focussed recharge is only around ~40% of the transmission losses.

Finally, considering the combination of explicit solutions of surface and subsurface components, the parsimonious structure, and the low computational cost, it is possible for DRYP to perform long runs using hourly or sub-hourly time steps. These characteristics enable DRYP to test long-term and seasonal changes in water availability to plants and humans in limited water environments under different scenarios and future climatic conditions such as anthropogenic activities or during droughts. Additionally, DRYP can be used in conjunction with stochastic rainfall simulation tools (such as STORM, Singer et al. (2018)) to explore the impact of the variability of precipitation on the water balance.

**Acknowledgements**
EAQ gratefully acknowledges financial support from Cardiff University through a Vice Chancellor's Scholarship. MOC gratefully acknowledges funding for an Independent Research Fellowship from the UK Natural Environment Research Council (NE/P017819/1). MBS acknowledges funding from the U.S. National Science Foundation (BCS-1660490, EAR-1700517) and the U.S. Department of Defense's Strategic Environmental Research and Development Program (RC18-1006). The team also acknowledge funding from the Global Challenges Research Fund (GCRF) ('Impacts of Climate Change on the Water Balance in East African Drylands'); The Royal Society ('DRIER', CHL\R1\180485); and the European Union's Horizon 2020 Programme ('DOWN2EARTH', 869550).

**Code availability.**
DRYP is available for download at https://github.com/AndresQuichimbo/DRYP (last access: 26 Mar 2021)
**Data availability.**
Our dataset contains modified Copernicus Climate Change Service information[1981-present. The model data used in this research is publicly available from https://www.tucson.ars.ag.gov/dap/.
**Author contributions.**
EAQ, implementation, conceptualization and data analysis, MOC conceptualization and data analysis. MBS conceptualization and data analysis. KM conceptualization and data analysis. DH conceptualization, implementation, and data analysis. RR conceptualization and data analysis. The paper was written by EAQ with contributions from all co-authors.
**Competing interests.**
The authors declare that they have no conflict of interest.
**Disclaimer.**
Any use of trade, firm, or product names is for descriptive purposes only and does not imply endorsement by the Cardiff University.

**List of parameters and model variables**

| Parameter | Description | Dimension |
|---|---|---|
| $AET$ | Actual evapotranspiration | $[L\ T^{-1}]$ |
| $AET_{SZ}$ | Capillary rise | $[L\ T^{-1}]$ |
| $B$ | Initial suction head | $[L]$ |
| $C$ | River conductivity | $[L^2\ T^{-1}]$ |
| $c$ | Readily available water factor | $[-]$ |
| $D_{root}$ | Rooting depth | $[L]$ |
| $D_{uz}$ | Unsaturated zone thickenss | $[L]$ |
| $ET_0$ | Reference evapotranspiration | $[L\ T^{-1}]$ |





| $f$ | Infiltration rate | [L T$^{-1}$] |
|---|---|---|
| $f_D$ | Effective aquifer depth (for exponential function) | [L] |
| $h$ | Water table elevation | [L] |
| $h_b$ | Aquifer bottom elevation | [L] |
| $h_{riv}$ | River stage elevation | [L] |
| $I$ | Cumulative infiltration | [L] |
| $I_c$ | Cumulative infiltration capacity | [L] |
| $ich$ | Channel losses | [L$^3$ T$^{-1}$] |
| $k$ | Crop coefficient | [-] |
| $K_{aq}$ | Aquifer Saturated hydraulic conductivity | [L T$^{-1}$] |
| $K_{ch}$ | Channel saturated hydraulic conductivity | [L T$^{-1}$] |
| $kdt$ | Schaake reference parameter | [-] |
| $K_{dtref}$ | Reference hydraulic conductivity | [L T$^{-1}$] |
| $K_{sat}$ | Saturated hydraulic conductivity | [L T$^{-1}$] |
| $k_T$ | Recession time for channel streamflow | [T$^{-1}$] |
| $L$ | Water thickness in the unsaturated zone | [L] |
| $L_{ch}$ | Channel length | [L] |
| $P$ | Precipitation | [L] |
| $p$ | Precipitation rate | [L T$^{-1}$] |
| $PAET_{SZ}$ | Maximum water uptake from saturated zone | [L T$^{-1}$] |
| $PET$ | Potential evapotranspiration | [L T$^{-1}$] |
| $q_0$ | Initial volumetric flow rate | [L$^3$ T$^{-1}$] |
| $Q_{ASW}$ | Surface water abstraction | [L$^3$] |
| $Q_{ASZ}$ | Groundwater abstraction | [L T$^{-1}$] |
| $Q_{in}$ | Channel inflow | [L$^3$] |
| $q_{in}$ | Volumetric flow rate entering stream cell | [L$^3$ T$^{-1}$] |
| $Q_{out}$ | Channel outflow | [L$^3$] |
| $q_{rir}$ | Saturation excess overland flow | [L T$^{-1}$] |
| $q_s$ | Groundwater discharge into streams | [L T$^{-1}$] |
| $Q_{TL}$ | Transmission losses | [L$^3$] |
| $R$ | Groundwater recharge | [L T$^{-1}$] |
| $r$ | Regularisation factor | [-] |
| $S_p$ | Sorptivity | [L$^2$ T$^{1/2}$] |
| $S_{SW}$ | Channel storage | [L$^3$] |
| $S_{SZ}$ | Storage in the saturated zone | [L] |
| $S_{UZ}$ | Storage in the unsaturated zone | [L] |
| $S_y$ | Specific yield | [-] |
| $T$ | Aquifer Transmissivity | [L$^2$ T$^{-1}$] |
| $TAW$ | Total available water for plant evapotranspiration | [L] |
| $W$ | Channel width | [L] |
| $z$ | Surface elevation | [L] |
| $z_{riv}$ | Bottom channel elevation | [L] |
| $\beta$ | Water stress coefficient | [-] |
| $\theta_{fc}$ | Water content at field capacity | [-] |
| $\theta_{sat}$ | Saturated water content | [-] |
| $\theta_{wp}$ | Water content at wilting point | [-] |
| $\lambda$ | Soil pore size distribution | [-] |





| $\mu_Y$ | Log mean saturated hydraulic conductivity | [L T$^{-1}$] |
| $\sigma_Y$ | Standard deviation of the log saturated hydraulic | [L T$^{-1}$] |
| $\psi_a$ | Initial suction head | [L] |
| $\psi_f$ | Suction head | [L] |

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






**Tables**

**Table 1.** Model parameters for different processes considered in the model, some required parameters depend on the infiltration approach ('Inf. Method'). Default values are specified in brackets. For soil hydraulic properties, default values correspond to a sandy loam soil texture (Clapp and Hornberger, 1978; Rawls et al., 1982).

| Parameter | Description | Dimension | Default Values | Inf. Method |
|---|---|---|---|---|
| **Overland flow** | | | | |
| $k_T$ | Recession time for channel streamflow | $[T^{-1}]$ | 0.083 h⁻¹* | - |
| $W$ | Channel width | $[L]$ | 10 m | - |
| $L_{ch}$ | Channel length | $[L]$ | grid size | - |
| $K_{ch}$ | Channel saturated hydraulic conductivity | $[L\,T^{-1}]$ | 10.9 mm h⁻¹ | - |
| **Unsaturated zone** | | | | |
| $\theta_{wp}$ | Water content at wilting point | $[-]$ | 0.07 | All |
| $\theta_{fc}$ | Water content at field capacity | $[-]$ | 0.17 | All |
| $\theta_{sat}$ | Saturated water content | $[-]$ | 0.41 | All |
| $\psi$ | Suction head | $[L]$ | 110.1 mm | All |
| $\lambda$ | Soil pore size distribution | $[-]$ | 4.9 | All |
| $\sigma_Y$ | Standard deviation of the log saturated hydraulic | $[LT^{-1}]$ | 0.5 mm h⁻¹ | Up-GA |
| $K_{sat}$ | Saturated hydraulic conductivity | $[L\,T^{-1}]$ | 120.9 mm h⁻¹ | All |
| $D$ | Rooting depth | $[L]$ | 800 mm | All |
| $kdt$ | Schaake reference parameter | $[-]$ | 1.0 | Schaake |
| $k$ | Crop coefficient | $[-]$ | 1.0 | - |
| **Saturated Zone** | | | | |
| $S_y$ | Specific yield | $[-]$ | 0.01 | - |
| $K_{aq}$ | Aquifer Saturated hydraulic conductivity | $[L\,T^{-1}]$ | 1 m h⁻¹ | - |
| $T$ | Aquifer Transmissivity (for constant values) | $[L\,T^{-1}]$ | 60 m² h⁻¹ | - |
| $f_D$ | Effective aquifer depth (for exponential function) | $[L]$ | 60 m | - |
| $h_b$ | Aquifer bottom elevation | $[L]$ | 0 m | - |

*Default values correspond to a flow velocity of ~1 m s⁻¹ over a 300-m straight path









**Figures**

**Figure 1.** Schematic representation of **DRYP** showing a) the main hydrological processes controlling water partitioning in
dryland regions; b) distributed datasets needed to derive input parameters; c) vertical and horizontal discretization and
representation of topographically-driven surface runoff, vertical flow in the unsaturated zone, and hydraulic gradient driven
groundwater flow in the saturated component; d) model structure and potential processes within a single grid cell for the
surface component (see Sect. 2.2), unsaturated zone (see Sect. 2.3) and saturated zone (see Sect. 2.4). Arrows represent flow
directions and red lines represent anthropogenic fluxes.


**Figure 2.** Schematic illustration of the unsaturated component. The right panel represents the variation of the ratio of potential
to actual evapotranspiration in relation to the water content of the soil. Please refer to Sect. 2.2 and 2.3 for a detailed explanation
of the terms shown here.

**Figure 3.** Schematic representation of a) UZ-SZ interactions: 1a) indicates no UZ-SZ interaction whereas 2a) indicates UZ-
SZ interaction (soil depth, $D_{root}$, is reduced to $D_{uz}$); b) SW-GW interactions in stream cells: boundary conditions change from
no-flow to head dependent flux conditions once the stream bed or ground surface is intersected by the water table. Upper part
of panel b) show the numerical implementation of SW-GW interactions in a stream cell.

**Figure 4.** Synthetic tilted-V catchment and flow boundary conditions specified for model simulations.

**Figure 5.** Geographic location of Walnut Gulch Experimental Watershed and location of monitoring stations

**Figure 6.** Temporal variation of a) precipitation (black line) and evapotranspiration (grey line), b) water content of the
unsaturated zone, c) groundwater recharge, d) runoff/discharge, and e) water table elevations. Right panels represent zoomed-
in sections of the shaded areas of the left panels. Solid lines represent the variation at the catchment outlet, whereas dashed
lines represent the temporal variation in the stream at 4 km from the catchment outlet. For panels b-e, blue lines represent the
'infiltration-discharge' scenario, green lines represent the 'Infiltration-evapotranspiration-discharge' scenario, and red lines
represent 'Infiltration-runoff-evapotranspiration-discharge' scenario.


**Figure 7.** Cumulative volume of main components of the water balance for the simulated scenarios: a) Infiltration – discharge,
b) infiltration - evapotranspiration - discharge, and c) infiltration-infiltration excess-evapotranspiration-discharge. P is the
precipitation, R is recharge, Q is discharge at the catchment outlet, AET is actual evapotranspiration, GWS is the change in
groundwater storage, and Err is the water balance error of the simulation.


**Figure 8.** Spatio-temporal visualisation of model process simulation at WGEW, a) rainfall event, b) soil moisture previous to
the rainfall event, c) ephemeral stream for the rainfall event, and c) soil moisture after the rainfall event; x and y axes distance
units are in metres.

**Figure 9.** Comparison between observed and simulated values of monthly temporal variation (left) and monthly distribution
(right) of a) monthly precipitation (left axes) and yearly precipitation (right axes), b) soil moisture at the COSMOS Kendall
location, c) actual evapotranspiration at Kendall, d) streamflow at flume F06, e) streamflow at flume F02, and f) streamflow
at flume F01. See Fig. 4 for station locations.

**Figure 10.** Average fluxes of different component of the water budget of WGEW for the simulated period, between 01/01/2007
and 01/01/2018. Blue arrows show input fluxes, green arrows represent water leaving the catchment, orange arrows represent
internal surface and unsaturated zone fluxes, and yellow arrows represent water moving to the saturated zone (not modelled).






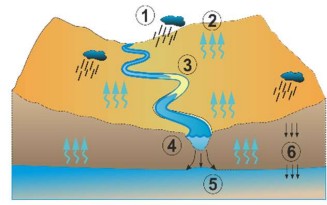

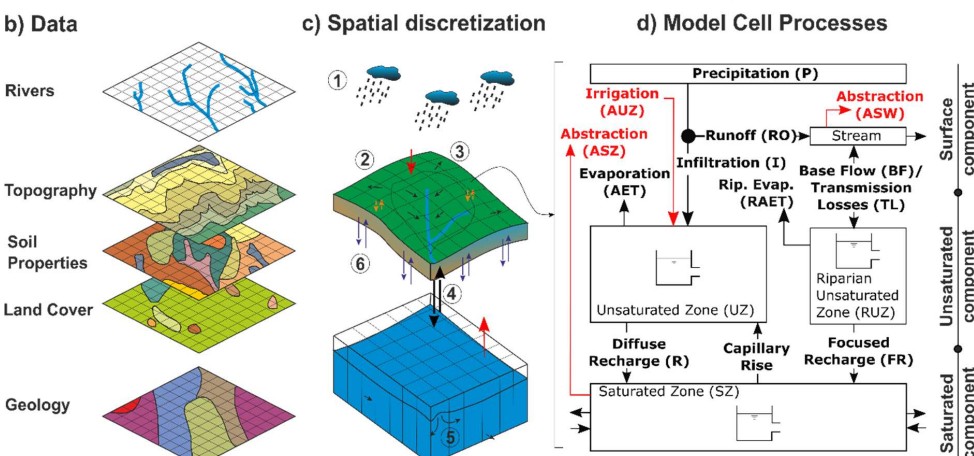

**Figure 1.** Schematic representation of **DRYP** showing a) the main hydrological processes controlling water partitioning in dryland regions; b) distributed datasets needed to derive input parameters; c) vertical and horizontal discretization and representation of topographically-driven surface runoff, vertical flow in the unsaturated zone, and hydraulic gradient driven groundwater flow in the saturated component; d) model structure and potential processes within a single grid cell for the surface component (see Sect. 2.2), unsaturated zone (see Sect. 2.3) and saturated zone (see Sect. 2.4). Arrows represent flow directions and red lines represent anthropogenic fluxes.


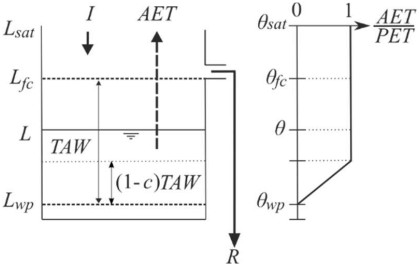

**Figure 2.** Schematic illustration of the unsaturated component. The right panel represents the variation of the ratio of potential to actual evapotranspiration in relation to the water content of the soil. Please refer to Sect. 2.2 and 2.3 for a detailed explanation of the terms shown here.




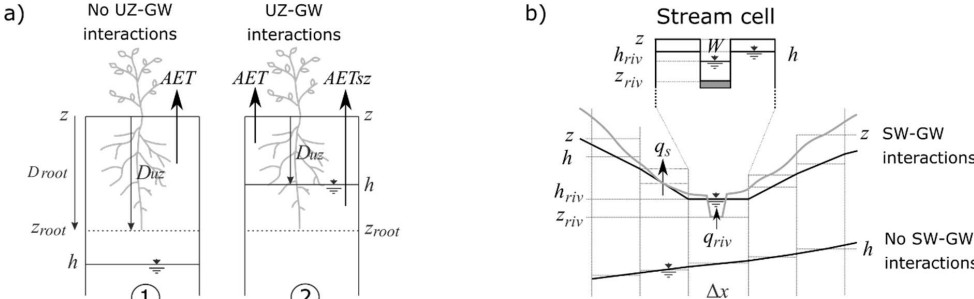

**Figure 3.** Schematic representation of a) UZ-SZ interactions: 1a) indicates no UZ-SZ interaction whereas 2a) indicates UZ-SZ interaction (soil depth, $D_{root}$, is reduced to $D_{uz}$); b) SW-GW interactions in stream cells: boundary conditions change from no-flow to head dependent flux conditions once the stream bed or ground surface is intersected by the water table. Upper part of panel b) show the numerical implementation of SW-GW interactions in a stream cell.

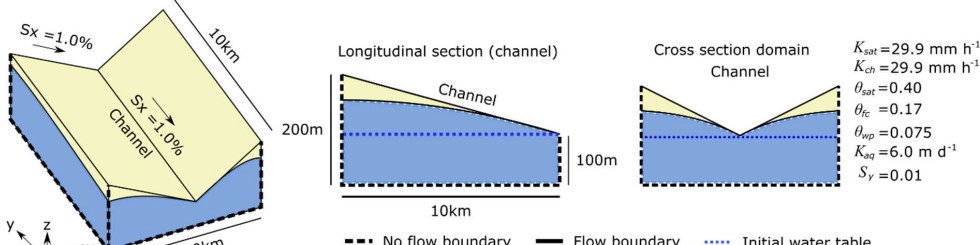

**Figure 4.** Synthetic tilted-V catchment and flow boundary conditions specified for model simulations.


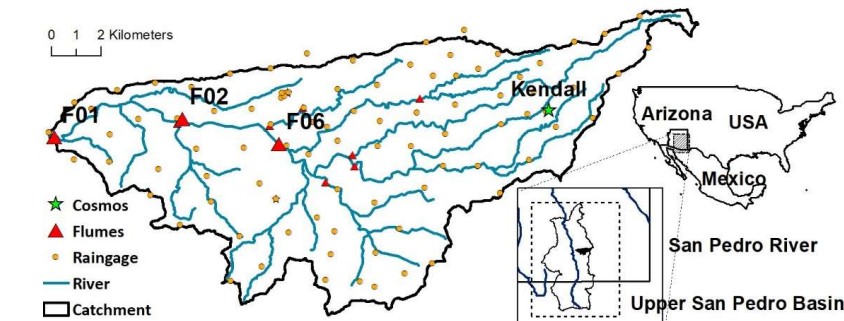

**Figure 5.** Geographic location of Walnut Gulch Experimental Watershed and location of monitoring stations



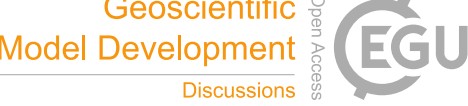
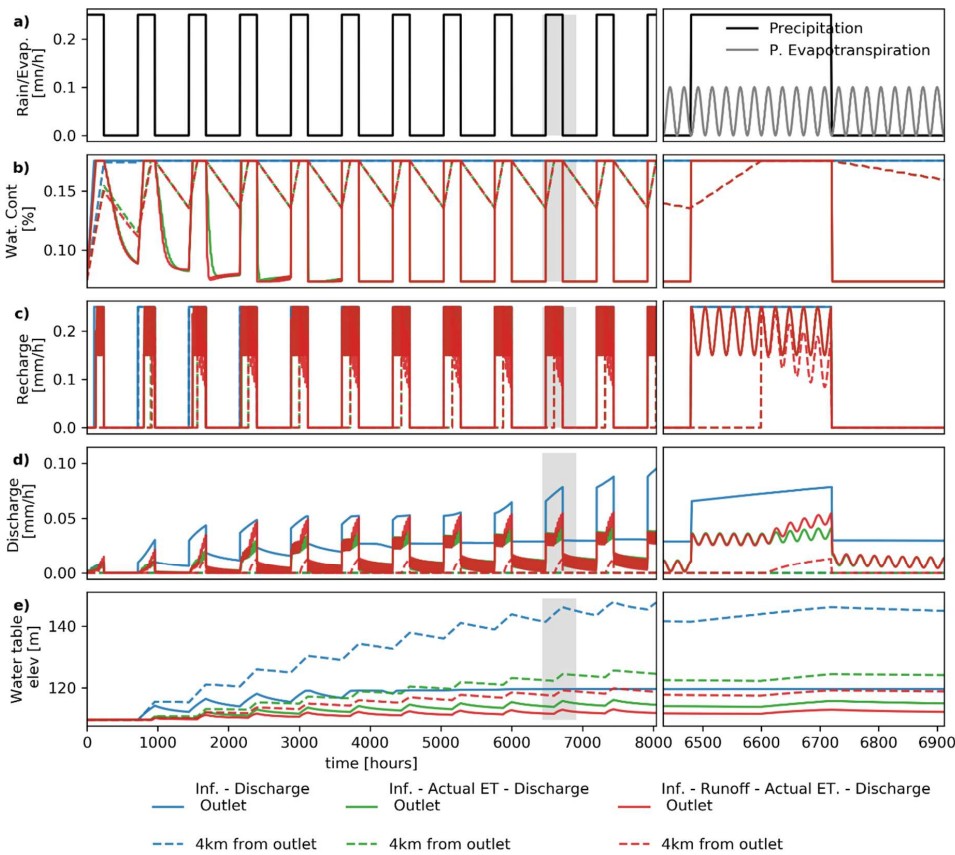

**Figure 6.** Temporal variation of a) precipitation (black line) and evapotranspiration (grey line), b) water content of the unsaturated zone, c) groundwater recharge, d) runoff/discharge, and e) water table elevations. Right panels represent zoomed-in sections of the shaded areas of the left panels. Solid lines represent the variation at the catchment outlet, whereas dashed lines represent the temporal variation in the stream at 4 km from the catchment outlet. For panels b-e, blue lines represent the 'infiltration-discharge' scenario, green lines represent the 'Infiltration-evapotranspiration-discharge' scenario, and red lines represent 'Infiltration-runoff-evapotranspiration-discharge' scenario.

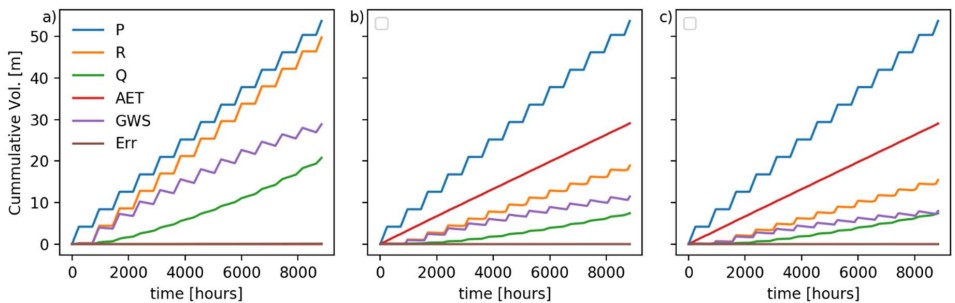

**Figure 7.** Cumulative volume of main components of the water balance for the simulated scenarios: a) Infiltration – discharge, b) infiltration - evapotranspiration - discharge, and c) infiltration-infiltration excess-evapotranspiration-discharge. P is the precipitation, R is recharge, Q is discharge at the catchment outlet, AET is actual evapotranspiration, GWS is the change in groundwater storage, and Err is the water balance error of the simulation.

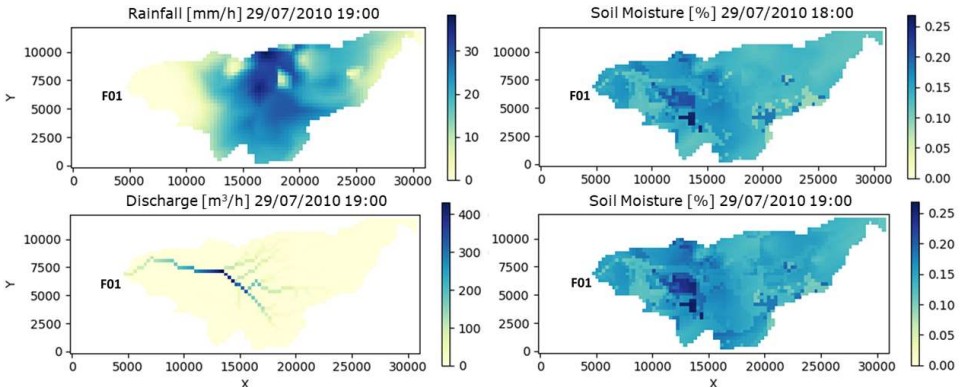

**Figure 8.** Spatio-temporal visualisation of model process simulation at WGEW, a) rainfall event, b) soil moisture previous to the rainfall event, and c) ephemeral stream for the rainfall event, and c) soil moisture after the rainfall event; x and y axes distance units are in metres.



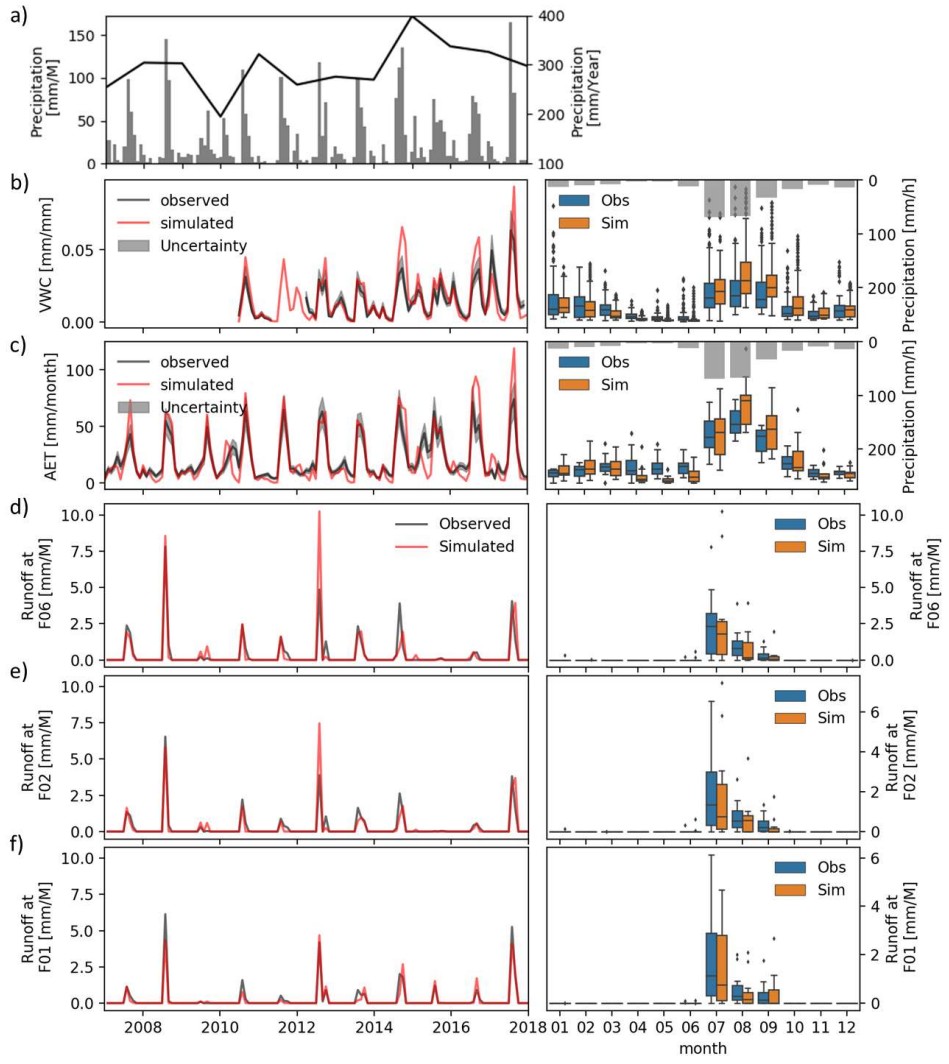

**Figure 9.** Comparison between observed and simulated values of monthly temporal variation (left) and monthly distribution (right) of a) monthly precipitation (left axes) and yearly precipitation (right axes), b) soil moisture at the COSMOS Kendall location, c) actual evapotranspiration at Kendall, d) streamflow at flume F06, e) streamflow at flume F02, and f) streamflow at flume F01. See Fig. 4 for station locations.

1400





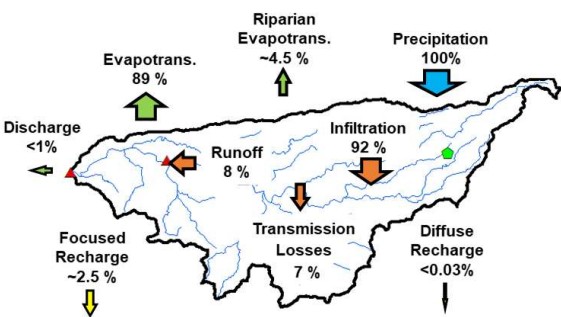

**Figure 10.** Average fluxes of different component of the water budget of WGEW for the simulated period, between 01/01/2007 and 01/01/2018. Blue arrows show input fluxes, green arrows represent water leaving the catchment, orange arrows represent internal surface and unsaturated zone fluxes, and yellow arrows represent water moving to the saturated zone (not modelled).