# Peer review of "DRYP 1.0: A parsimonious hydrological model of DRYland Partitioning of the water balance"

_Geoscientific Model Development, 2021_

## Author Comment (AC1)

**Comment 1.** The authors have brought together many analyses into a useful, integrated platform. As they point out, a fully distributed model that can capture all of the processes represented would be unwieldy at best and unattainable at worst. On this basis, I fully support the development of this tool.

**Response 1.** We thank Prof Ferre for taking the time to review the article, being supportive of the aims of our modelling endeavour, and for providing very helpful comments that will certainly improve the manuscript.

**Comment 2.** My only concern is that this model, while an improvement, still requires many simplifications. The authors have sufficient experience and expertise to know which module is best to use for specific conditions ... especially as they are familiar with the Walnut Gulch site and with modeling COSMOS data. Could they provide more guidance for less well-informed users? Which assumptions are most critical? Under what ranges of conditions should this model not be used? How could a user identify potential systematic errors from the model results? In short, simpler models are better - but, only if they aren't significantly flawed. Can the authors help to make sure that their model will be used appropriately and constructively?

**Response 2.** Thank you for raising this concern – the model does indeed require many simplifications, and we agree that it would be useful for us to add more to the revised paper to help make sure the model will be used appropriately.

In terms of adding more guidance for users, we will re-iterate that one of the main advantages of parsimonious models is the ability to be efficiently (numerically/computationally) implemented. This allows more robust evaluation of the uncertainty of the model by considering combined structural and parameter uncertainty estimation, for example via Monte Carlo experiments. Such uncertainty analysis, which is made possible by DRYP's relative parsimony, can give insights into the potential systematic errors resulting from the model, and we will make it clearer in the revised paper that users are strongly advised to perform such analyses to ensure that the model is fit for purpose for their intended application.

In terms of the ranges of conditions for which the model is not applicable, we note that overland flow and routing processes in dryland regions are often highly dynamic mechanisms, both temporally and spatially, and greatly influence the water partitioning. DRYP has not been developed to capture these characteristics, which may be required for other applications such as flood forecasting. Rather, we aim to capture the long-term changes/processes of the main components of the hydrological cycle, such as drought, as stated in Lines 99-101: "*We do not intend for this model to accurately simulate event-based flood hydrographs, for example, for flood hazard analysis. Instead, we aimed to develop a model that captures the long-term behaviour of the water balance in dryland regions.*" Hence, in this context, an important way the model should *not* be used is if the user is looking to simulate the highly dynamic variation of flow in channels or on the surface. In more general terms, we don't see any particular restriction on the use of the model for other climatic conditions beyond drylands, as long as the main assumptions, especially regarding the infiltration and flow routing components, are considered valid to the particular application. We will update the revised manuscript to discuss these aspects in more detail.

**Comment 3:** If I have one other quibble, it is with the testing of the model. It is true that spatially distributed numerical models are expensive to run and I wouldn't suggest that they need to build and

run one to test the applicability of their simplified model. But, I would have thought that someone, somewhere, would have a model that could be used as a basis for comparison to give a better estimate of the magnitude and distribution of errors. It isn't my field, exactly, but I'm not sure that mass balance errors alone are sufficient to validate a model.

**Response 3**: Thanks for this suggestion. In response, we have now run some additional synthetic experiments against the industry standard software MODFLOW, to support the documentation of the ability of DRYP to represent the surface-water groundwater process interactions robustly. The results show excellent agreement between DRYP and MODFLOW models with respect to how the groundwater components interact with the drainage, as can be seen in the summary of the results shown below, and these experiments will be documented in detail in the revised paper.

[Figure]

a) Simulated head along the aquifer for different time steps estimated by DRYP (solid lines) and MODFLOW (dashed lines), and b) temporal variation of the mass balance error for DRYP.

**Comment 4**. All of these comments should be seen as suggestions only. It is entirely possible that I am out of touch with what is expected/required for models at this scale and especially for use in hydrometeorology. I leave it entirely to the authors' discretion to decide to consider or ignore my comments. Nice work! Ty Ferre

P.S. I did see one spelling error on a plot axis ... cummulative. I first thought that it might be a British spelling, but Google doesn't support that hypothesis.

**Response 4.** Many thanks – your comments were very useful and we hope you feel we have done them justice in our responses. We will correct the spelling error you have correctly spotted in the revised paper.

---

## Author Comment (AC3)

**Comment 1.** Authors have developed a distributed parsimonious hydrologic model to represent surface water-groundwater interactions and channel transmission losses in dryland environments. The modeling package is written in Python and provides some flexibility in terms of implementing a variable time step for different hydrologic processes and using different equations to represent infiltration. The new modeling package was set-up over the Walnut Gulch experimental watershed in Arizona and results were validated using observed streamflow, soil moisture and actual evapotranspiration data. Despite a relatively simple structure, the model was able to capture the main hydrologic processes of the studied catchment. The paper is well written and authors have performed robust model evaluation using synthetic and real numerical experiments. I provide a few comments below to further clarify some of the points discussed in the paper.

**Response 1.** We greatly thank the reviewer for taking the time to review the article and provide useful comments that will certainly improve the manuscript. We have addressed your comments in the text below.

**Comment 2:** Line 75- Could you please explain which "hard-coded parameterization" you refer to here.

**Response 2:** We refer here to any physical and numerical parameter used for ensuring the stability and convergence of numerical solutions, particularly when using highly non-linear approaches such as the Richard's equation. We will clarify this statement in the revised version of the manuscript.

**Comment 3:** Line 145- How do you represent the streams? Is width of a stream equal to the width of a grid cell or do you have flexibility in representing the streams using different geometries? While this information is added later in the text, it would be more useful to state it earlier in the manuscript.

**Response 3:** Stream width and length can vary if the user provides these values. We have restricted the stream cross-sections to rectangular shapes to reduce the computational demand when using cross-sections with different shapes. A simple cross-section also prevents the use of an implicit scheme to solve water partitioning on the channel.

We will make these points clearer in the revised manuscript.

**Comment 4:** Line 335- Do you assume the subsurface is homogenous for both saturated and unsaturated zones?

**Response 4:** Yes, it is assumed that each compartment is homogenous, however, each compartment has its own hydraulic properties.

**Comment 5:** In Equation 20- Do you assume a constant specific yield across the entire subsurface?

**Response 5:** The specific yield is constant for each cell for saturated conditions. For unsaturated conditions, storage is controlled by soil storage capacity. Therefore, when the water table is below the rooting depth the model uses Sy, but when the water table is above the rooting depth elevation, the soil storage capacity is considered in the model. It is described in section 4.1, but we will clarify it in the revised manuscript.

**Comment 6:** How do you define the riparian unsaturated zone? does the extent of the riparian zone variable or fixed?

**Response 6:** The riparian zone uses a similar approach as the soil unsaturated zone. It is defined as a region parallel to the stream, and its extent can be specified by using a raster file. A value of 20m has

been specified as a default condition. The hydraulic conductivity in the riparian zone is specified by the hydraulic conductivity of the channel streambed.

We will clarify the riparian zone characteristics in the corresponding section of the revised manuscript and the user control which is possible for this aspect.

**Comment 7:** Do you implement an iterative scheme to consider the two-way coupling between the saturated and unsaturated zones? Please clarify.

**Response 7:** The model uses a non-iterative scheme as described in section 2.1, lines 132-135. We have implemented a non-iterative scheme in order to reduce the computational demand that is required when using iterative schemes. We will make sure that it is clearly stated in the corresponding section in the revised manuscript.

**Comment 8:** I suggest to move section 2.6 after section 2.1. to first describe the model inputs and then explain the equations.

**Response 8:** Thanks for the suggestion, we will move section 2.6 to the place that helps the reader to better understand the manuscript.

**Comment 9:** What are the vegetation specific parameters in the model? It seems only the rooting depth is set depending on the landcover type.

**Response 9:** Three parameters control the vegetation in the model: the rooting depth, the crop factor $k$, and water stress fraction, $c$. It is currently described in section 2.3, lines 360-378. However, we will make sure that it is better described in the revised paper.

**Comment 10:** Line 635- Since the model resolution is 300 m, it means the width of the stream cell is 300 m as well. This width is too large to represent the width of streams in arid regions.

**Response 10:** As noted above the stream width and length can be set by the user in each cell of the model. Here, stream width is specified as 10m based on the average channel width of the main channels of the WGEW. We will make sure that it is better described in the revised version of the paper.

**Comment 11:** Figure 9- Why do you show uncertainty bounds for the simulated soil moisture and AET and not the runoff? does the "red" simulated line represent the best model run in Figure 9?

**Response 11:** We will add the uncertainty band for the streamflow figures. Regarding the plotted values, yes, we are showing results for the best simulation. We will clarify it in the corresponding section of the revised paper as well as in the figure caption.

**Comment 12:** Please discuss the limitations of the current model and future development plans.

**Response 12:**  We will extend the discussion section to include future model development plans. In short, we will improve the soil-vegetation interaction in the unsaturated zone to capture the temporal variation of plant water demand. We are planning to extend the model to regional scales. We are also looking at the integration of the model with weather stochastic tools to enhance the ability of the model to test long-term variations of climatic conditions on water partitioning.

Minor Comments:

**Comment 13**. Line 21- Add "distributed" to the model to provide this important information to the reader upfront.

Many thanks for the suggestion, we will add it in the revised version.

**Comment 14**. Line 118 – Remove "(a la" from the reference

Thanks, we will remove it.

**Comment 15**. Line 120- Replace "streambed" with "stream stage".

Thanks for finding this error, we will change it in the revised version.

**Comment 16**. Line 131 – Replace "an" with "a"

 Thanks, we will change it.

**Comment 17**. Line 275 – Add "runoff" to the next downstream cell

Thanks, we will add it to the revised version.

**Comment 18**. Line 355- Replace "relative water content" with  "volumetric water content" .

Thanks, we will change it.

**Comment 19**. Line 823 – Change "Fig. 9a" to "Fig. 9b".

Thanks, we will fix it in the revised version.

**Comment 20**. Line 832 – Change "Fig. 9b" to "Fig. 9c".

Thanks, we will fix it in the revised version.

**Comment 21**. Line 841- Change "Fig. 9c" to "Fig. 9f".

Thanks, we will fix it in the revised version.

---

## Author Response (AR1)

**Response to reviewers:**

**Paper:** "DRYP 1.0: A parsimonious hydrological model of DRYland Partitioning of the water balance" by Quichimbo E. A. et al., Geosci. Model Dev. Discuss., https://doi.org/10.5194/gmd-2021-137-AC1, 2021

We greatly thank the reviewers for taking the time for reading this manuscript, their comments and suggestions have greatly enhanced the quality of this manuscript. We have addressed all your comments below (our responses are in red):

**Response to Prof Ferre**

**Comment 1**. The authors have brought together many analyses into a useful, integrated platform. As they point out, a fully distributed model that can capture all of the processes represented would be unwieldy at best and unattainable at worst. On this basis, I fully support the development of this tool.

**Response 1**. We thank Prof Ferre for taking the time to review the article, being supportive of the aims of our modelling endeavour, and for providing very helpful comments that will certainly improve the manuscript.

**Comment 2**. My only concern is that this model, while an improvement, still requires many simplifications. The authors have sufficient experience and expertise to know which module is best to use for specific conditions ... especially as they are familiar with the Walnut Gulch site and with modeling COSMOS data. Could they provide more guidance for less well-informed users? Which assumptions are most critical? Under what ranges of conditions should this model not be used? How could a user identify potential systematic errors from the model results? In short, simpler models are better - but, only if they aren't significantly flawed. Can the authors help to make sure that their model will be used appropriately and constructively?

**Response 2.** Thank you for raising this concern – the model does indeed require many simplifications, and we agree that it would be useful for us to add more to the revised paper to help make sure the model will be used appropriately.

In terms of adding more guidance for users, we will re-iterate that one of the main advantages of parsimonious models is the ability to be efficiently (numerically/computationally) implemented. This allows more robust evaluation of the uncertainty of the model by considering combined structural and parameter uncertainty estimation, for example via Monte Carlo experiments. Such uncertainty analysis, which is made possible by DRYP's relative parsimony, can give insights into the potential systematic errors resulting from the model, and we will make it clearer in the revised paper that users are strongly advised to perform such analyses to ensure that the model is fit for purpose for their intended application.

We appreciate that it being computationally efficient does not necessarily mean that the model is parsimonious (e.g., complex hydrological or land surface models running in high-performance computers). We believe the parsimony aspect of DRYP comes from the fact that the model structure has been designed to be as simple as possible, based on our expert knowledge (as well as from our literature review), to better represent hydrological stores and fluxes in drylands. This allows one to better identify mechanisms and controlling factors in those regions in which traditional (more general)

hydrological models may not be able to capture them appropriately (e.g., the role of focused versus diffuse discharge). We have added the following text on lines 929 – 933 in the discussion section to address your comment:

*"DRYP is parsimonious in that the model structure is has few tunable parameters, but still captures the essential elements of dryland hydrology and well represents hydrological stores and fluxes in drylands. It is designed to allow the user to identify mechanisms and factors that affect the water balance in dryland regions, where most existing hydrological models may not be able to capture them appropriately (e.g., the role of focused versus diffuse discharge)"*

In terms of the ranges of conditions for which the model is not applicable, we note that overland flow and routing processes in dryland regions are often highly dynamic mechanisms, both temporally and spatially, and greatly influence the water partitioning. The current version of DRYP has not been developed to capture these characteristics, which may be required for other applications such as flood forecasting. In this version, we aimed to capture the long-term changes/processes of the main components of the hydrological cycle, such as drought, as stated in Lines 99-101: *"We do not intend for this model to accurately simulate event-based flood hydrographs, for example, for flood hazard analysis. Instead, we aimed to develop a model that captures the long-term behaviour of the water balance in dryland regions."* Hence, in this context, we note that the model should *not* be used to simulate the highly dynamic variation of flow in channels or on the surface. In more general terms, we don't see any particular restrictions on the use of the model for other climatic conditions beyond drylands, as long as the main assumptions regarding the infiltration and flow routing components, are considered valid for the particular application. We have extended the discussion sections to clarify the limitations of the model, for example, as listed in the response to Comment 12 of reviewer 2 below.

**Comment 3**: If I have one other quibble, it is with the testing of the model. It is true that spatially distributed numerical models are expensive to run and I wouldn't suggest that they need to build and run one to test the applicability of their simplified model. But, I would have thought that someone, somewhere, would have a model that could be used as a basis for comparison to give a better estimate of the magnitude and distribution of errors. It isn't my field, exactly, but I'm not sure that mass balance errors alone are sufficient to validate a model.

**Response 3**: Thanks for this suggestion. In response, we have now run some additional synthetic experiments against the industry standard software MODFLOW, to support the documentation of the ability of DRYP to represent the surface-water groundwater process interactions robustly. The results show excellent agreement between DRYP and MODFLOW models with respect to how the groundwater components interact with the drainage. We have documented these results and added the following text in the evaluation section, lines 573-696:

"

Hence, here we have considered two sets of model evaluations: (i) a quantitative evaluation of the model performance in relation to the well know numerical model, MODFLOW, for a simple surface-groundwater interaction test represented as a draining condition, and (ii) a qualitative evaluation of the model performance with respect to the desired skill of the model to seamlessly allow interactions between groundwater and the land surface and surface water components.

*(i) Comparing DRYP and MODFLOW*

For the quantitative evaluation, a 1-D synthetic experiment considering an inclined plane aquifer was set-up using DRYP (see Fig 4a). The length and width of the model domain were specified as 10 km and 1 km, respectively.

Hydraulic saturated conductivity and aquifer specific yield were specified as 1.2 m d$^{-1}$ and 0.01, respectively. Boundary conditions were specified as no-flow for both the right and left side as well as the bottom of the model domain. The model grid size was set to 1 km × 1km.

A model with identical geometry, grid size and hydraulic properties was built in MODFLOW using the FLOPY python package (Bakker et al., 2016b, a). Boundary conditions for the MODFLOW model were the same as DRYP except for the top boundary condition, which was specified using the drain package (Harbaugh et al., 2000). The elevation at which the water starts to drain was specified as the top surface elevation of the model domain. A high value of conductivity term (500 m$^2$ d$^{-1}$) was used in order to capture the seepage process and to assure convergence as well as minimal water balance errors (Batelaan and Smedt, 2004).

The synthetic test consisted of a free-draining condition for an unconfined aquifer with a water table depth equal to zero (at the surface level). The time step used for evaluation was 1 day. The evaluation considered the temporal variation of the water table for both DRYP and MODFLOW models, as well as the water balance errors. Errors were evaluated at all locations along the aquifer. Mass balance errors were estimated by the algebraic sum of inputs, outputs and the storage change.

*(ii) Qualitative analysis of surface groundwater-interactions*

The geometry of the model domain for the qualitative tests consisted of a tilted-V catchment (Fig. 4) with a size of 7×10 square cells on a 1-km resolution grid.

[Figure]

Figure 6. a) Simulated head along the aquifer for different time steps (in months, M) estimated by DRYP (solid lines) and MODFLOW (dashed lines), and b) temporal variation of the mass balance error for DRYP
"

We have also modified figure 4 to include the setup of the new analysis:

[Figure]

**Figure 4.** Model domain for synthetic experiment: a) 1-D model, and b) tilted-V catchment and flow boundary conditions specified for model simulations.

**Comment 4**. All of these comments should be seen as suggestions only.  It is entirely possible that I am out of touch with what is expected/required for models at this scale and especially for use in hydrometeorology.  I leave it entirely to the authors' discretion to decide to consider or ignore my comments. Nice work! Ty Ferre

P.S. I did see one spelling error on a plot axis ... cummulative.  I first thought that it might be a British spelling, but Google doesn't support that hypothesis.

**Response 4.** Many thanks – your comments were very useful and we hope you feel we have done them justice in our responses. We have corrected the spelling error of figure 7.

**Response to reviewer #2**

**Comment 1.** Authors have developed a distributed parsimonious hydrologic model to represent surface water-groundwater interactions and channel transmission losses in dryland environments. The modeling package is written in Python and provides some flexibility in terms of implementing a variable time step for different hydrologic processes and using different equations to represent infiltration. The new modeling package was set-up over the Walnut Gulch experimental watershed in Arizona and results were validated using observed streamflow, soil moisture and actual evapotranspiration data. Despite a relatively simple structure, the model was able to capture the main hydrologic processes of the studied catchment. The paper is well written and authors have

performed robust model evaluation using synthetic and real numerical experiments. I provide a few comments below to further clarify some of the points discussed in the paper.

**Response 1.** We greatly thank the reviewer for taking the time to review the article and provide useful comments that will certainly improve the manuscript. We have addressed your comments in the text below.

**Comment 2:** Line 75- Could you please explain which "hard-coded parameterization" you refer to here.

**Response 2:** We refer here to any physical and numerical parameters used for ensuring the stability and convergence of numerical solutions, particularly when using highly non-linear approaches such as the Richards equation. We have added the following text to clarify this statement in line 77-78: *"required to ensure convergence and numerical stability"*.

**Comment 3:** Line 145- How do you represent the streams? Is width of a stream equal to the width of a grid cell or do you have flexibility in representing the streams using different geometries? While this information is added later in the text, it would be more useful to state it earlier in the manuscript.

**Response 3:** Stream width and length can vary if the user provides these values. We have restricted the stream cross-sections to rectangular shapes to reduce the computational demand that would arise from using cross-sections with different shapes. A simple cross-section also enables the use of an explicit scheme to solve water partitioning on the channel.

We have added the following text to make these points clearer in lines 149-150: *"The size of the stream and riparian zone is only limited by the grid size."* And lines 301-302: *"In order to use an explicit approach at the same time as maintaining the simplicity of the model, the channel cross section is assumed to be rectangular."*

**Comment 4:** Line 335- Do you assume the subsurface is homogenous for both saturated and unsaturated zones?

**Response 4:** Yes, it is assumed that each compartment is homogenous. However, each compartment has its own hydraulic properties. This is described in section 2.4.1.

**Comment 5:** In Equation 20- Do you assume a constant specific yield across the entire subsurface?

**Response 5:** The specific yield is constant for each cell for saturated conditions. For unsaturated conditions, water storage is controlled by soil storage capacity. Therefore, when the water table is below the rooting depth the model uses $S_y$, but when the water table is above the rooting depth elevation, the soil storage capacity is computed explicitly in the model. This is described in section 2.4.1.

**Comment 6:** How do you define the riparian unsaturated zone? does the extent of the riparian zone variable or fixed?

**Response 6:** The riparian zone uses a similar approach as for the soil unsaturated zone. It is defined as a region parallel to the stream, and its extent can be specified by using a raster file. A value of 20 m has been specified as a default condition, but this can be adjusted. The hydraulic conductivity in the riparian zone is specified by the hydraulic conductivity of the channel streambed.

We have added the following text to clarify the parameters of the riparian zone in lines 422-455: *"The riparian zone uses, by default, the same hydraulic properties of the soil unsaturated zone except the saturated hydraulic conductivity which is assumed the same as the channel streambed Kch, however, these parameters are also user-defined. The size of the riparian zone has a user-defined width (default is 20 m) and the length is the same as the stream."*

**Comment 7:** Do you implement an iterative scheme to consider the two-way coupling between the saturated and unsaturated zones? Please clarify.

**Response 7:** The model uses a non-iterative scheme as described in section 2.1, lines 135-137. We have implemented a non-iterative coupling scheme in order to reduce the computational demand that is required when using iterative schemes.

**Comment 8:** I suggest to move section 2.6 after section 2.1. to first describe the model inputs and then explain the equations.

**Response 8:** Thanks for the suggestion, we have moved section 2.6 to section 2.2.

**Comment 9:** What are the vegetation specific parameters in the model? It seems only the rooting depth is set depending on the landcover type.

**Response 9:** Three parameters control the vegetation in the model: the rooting depth, the crop factor *k*, and water stress fraction, *c*. These parameters are described in section 2.3, lines 376-398.

**Comment 10:** Line 635- Since the model resolution is 300 m, it means the width of the stream cell is 300 m as well. This width is too large to represent the width of streams in arid regions.

**Response 10:** As noted above, the stream width and length can be set by the user in each cell of the model. Here, stream width is specified with a default value of 10 m, based on the average channel width of the main channels of the WGEW. We have added the following text in lines 672-673: *"Stream width was assumed as 10 m for the whole model domain based on average values over the whole catchment (Miller et al., 2000)."*

**Comment 11:** Figure 9- Why do you show uncertainty bounds for the simulated soil moisture and AET and not the runoff? does the "red" simulated line represent the best model run in Figure 9?

**Response 11:** We have added the uncertainty bounds for the streamflow figures. Regarding the plotted values, yes, we are showing results for the best simulation. We have modified the figure caption to reflect that change: *"Figure 9. Comparison between observed and the best simulated run of monthly temporal variation (left) and monthly distribution (right) of a) monthly precipitation (left axes) and yearly precipitation (right axes), b) soil moisture at the COSMOS Kendall location, c) actual evapotranspiration at Kendall, d) streamflow at flume F06, e) streamflow at flume F02, and f) streamflow at flume F01. See Fig. 4 for station locations."*

**Comment 12:** Please discuss the limitations of the current model and future development plans.

**Response 12:** We have extended the discussion section to include future model development plans. Specifically, the following text has been added at lines 953-965:

*"*

*Considering the combination of explicit solutions of surface and subsurface components, the parsimonious structure, and the low computational cost, it is possible for DRYP to perform long runs using hourly or sub-hourly time steps. These characteristics enable DRYP to test long-term and seasonal changes in water availability to plants and humans in limited water environments under different scenarios and future climatic conditions such as anthropogenic activities or during droughts. Additionally, given the minimal data requirements, DRYP has the potential to be applied in areas where only information at large scales is available.*

*Furthermore, improving the soil-vegetation interaction in the unsaturated zone to capture the temporal variation of plant water demand will likely enhance the performance of the model. A more complex representation of the highly dynamic behaviour of ephemeral streamflow will be considered in future developments in order to enhance the ability of the model to represent flooding conditions. Additionally, we also plan to use DRYP in conjunction with stochastic rainfall simulation tools (such as STORM, Singer et al. (2018)) to explore the impact of the variability of precipitation on the water balance..*"

Minor Comments:

**Comment 13**. Line 22- Add "distributed" to the model to provide this important information to the reader upfront.

Many thanks for the suggestion, we have added this term in the revised version.

**Comment 14**. Line 221 – Remove "(a la" from the reference

Thanks, we have removed it.

**Comment 15**. Line 123- Replace "streambed" with "stream stage".

Thanks for finding this error, we have changed it in the revised version.

**Comment 16**. Line 134 – Replace "an" with "a"

 Thanks, we have changed it.

**Comment 17**. Line 297 – Add "runoff" to the next downstream cell

Thanks, we have added it to the revised version.

**Comment 18**. Line 378- Replace "relative water content" with  "volumetric water content" .

Thanks, we have changed it.

**Comment 19**. Line 876 – Change "Fig. 9a" to "Fig. 9b".

Thanks, we have fixed it in the revised version.

**Comment 20**. Line 885 – Change "Fig. 9b" to "Fig. 9c".

Thanks, we have fixed it in the revised version.

**Comment 21**. Line 894- Change "Fig. 9c" to "Fig. 9f".

Thanks, we have fixed it in the revised version.